# K²-Agent: Co-Evolving Know-What and Know-How for Hierarchical Mobile Device Control

**Zhe Wu**[*1], **Donglin Mo**[*1], **Hongjin Lu**[1], **Junliang Xing**[†1]
**Jianheng Liu**[2], **Yuheng Jing**[3], **Kai Li**[3], **Kun Shao**[2], **Jianye Hao**[2], **Yuanchun Shi**[1]
[1]Department of Computer Science and Technology, Tsinghua University
[2]Huawei Noah's Ark Lab, [3]Institute of Automation, Chinese Academy of Sciences

## Abstract

Existing mobile device control agents often perform poorly when solving complex tasks requiring long-horizon planning and precise operations, typically due to a lack of relevant task experience or unfamiliarity with skill execution. We propose **K²-Agent**, a hierarchical framework that models human-like cognition by separating and co-evolving declarative ("*knowing what*") and procedural ("*knowing how*") knowledge for planning and execution. K²-Agent's high level reasoner is bootstrapped from a single demonstration per task and runs a Summarize–Reflect–Locate–Revise (SRLR) loop to distill and iteratively refine task-level declarative knowledge through self-evolution. The low-level executor is trained with our curriculum-guided Group Relative Policy Optimization (C-GRPO), which (i) constructs a balanced sample pool using decoupled reward signals and (ii) employs dynamic demonstration injection to guide the model in autonomously generating successful trajectories for training. On the challenging AndroidWorld benchmark, K²-Agent achieves a 76.1% success rate using only raw screenshots and open-source backbones. Furthermore, K²-Agent shows powerful dual generalization: its high-level declarative knowledge transfers across diverse base models, while its low-level procedural skills achieve competitive performance on unseen tasks in ScreenSpot-v2 and Android-in-the-Wild (AitW).

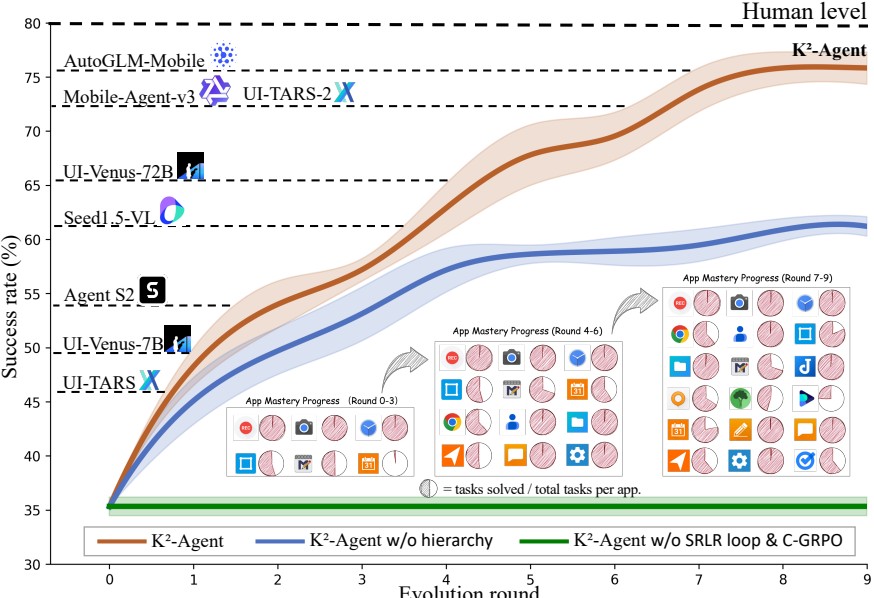

Figure 1: **K²-Agent's co-evolutionary learning curve on AndroidWorld.** The main curve shows the agent's success rate steadily improving. Ablations (lower curves) confirm the contribution of key components, and subplots below illustrate the expanding mastery over new apps and tasks.

[*]Equal contribution
[†]Corresponding Author: jlxing@tsinghua.edu.cn

# 1 INTRODUCTION

Human intelligence relies on a fundamental distinction between two types of knowledge (Squire, 2009): declarative knowledge (*knowing what*) and procedural knowledge (*knowing how*). Declarative knowledge is symbolic and explicit; it can be articulated, summarized from one or a few demonstrations, and refined through recall. In contrast, procedural knowledge manifests as executable skills (e.g., cycling, swimming). It is often implicit, difficult to verbalize, and acquired through repeated practice to form "muscle memory". These two knowledge systems, believed to be supported by distinct cognitive circuits (Squire & Knowlton, 1995), are co-activated in complex tasks—one to decide what to do, the other to determine how to do it.

This division of labor is especially critical for long-horizon mobile device control. Recall when you performed a multi-step task in an unfamiliar app. You need declarative knowledge about the task to guide the generation of operational intent. For example, you need to know in advance that clicking the "trash icon" means delete. At the same time, you need procedural knowledge to accurately translate that intent into atomic actions such as clicking, swiping, and text input.

Existing methods for mobile device control (Li et al., 2024) largely fall into two lines. (i) *Training-free agents* (Zhang et al., 2025; Wang et al., 2024b; Agashe et al., 2024; 2025). These agents carefully design workflows and encode task-related knowledge into prompts or in-context examples. Development is relatively cheap and edits are quick, but their performance is capped by the foundation models, which are often closed-source and cannot be fine-tuned to fix domain-specific, persistent errors. (ii) *Learning-based agents* (Hong et al., 2024; Pan et al., 2024; Zhang & Zhang, 2023; Bai et al., 2024; Wang et al., 2024c). These agents train parametric policies with supervised fine-tuning (SFT) or reinforcement learning (RL) on large labeled datasets. While stable on in-distribution actions, they struggle with long-horizon credit assignment and poor task generalization.

On the decision paradigm, recent work increasingly separates reasoning from action (Qin et al., 2025; Gu et al., 2025), or adopts an explicit planner–executor hierarchy (Agashe et al., 2024; 2025). In practice, this design often beats flat policies. However, most hierarchies remain only a structural split. Either both layers are training-free or both are trained with SFT/RL. This results in systems relying either on extensive manual design or on massive amounts of data and computational resources (typically requiring 10k+ samples and hundreds of GPUs) (Qin et al., 2025; Gu et al., 2025). **Our key insight is that *know-what* and *know-how* naturally match the hierarchical design; they should follow different update rules and co-evolve through continuous interaction.**

To this end, we propose **K²-Agent**, a hierarchical planner–executor framework for mobile device control. It explicitly decouples and co-evolves declarative (*know-what*) and procedural (*know-how*) capabilities, while connecting high-level planning and low-level execution through clear single-step sub-goals. The **high-level planner** starts from one demonstration per task and runs a Summarize–Reflect–Locate–Revise (SRLR) loop that keeps an updatable task memory. Using execution feedback collected in the loop, SRLR locates failure points and revises the knowledge so the plan improves over time. The **low-level executor** is trained using our curriculum-guided Group Relative Policy Optimization (C-GRPO). The method first decouples task execution rewards across action-type and parameter dimensions, routing samples into different experience pools by error type for proportional sampling. It then introduces a dynamic demonstration injection that prepends variable-length expert prefixes to the model's prompts, conditioned on sample difficulty and training stage. This guides the model to autonomously generate successful trajectories for GRPO-style training, thereby building a reusable skill library. These two evolutionary processes are mutually coupled, forming a closed-loop system where "thinking" and "practice" reinforce each other.

On the challenging AndroidWorld benchmark (Rawles et al., 2024), K²-Agent sets a new SOTA with a **76.1%** success rate. This surpasses all leading learning-based models (Qin et al., 2025; Gu et al., 2025; Ye et al., 2025) and rivals top closed-source models (FinalRun, 2025) that leverage additional inputs from the accessibility (A11y) tree. More importantly, K²-Agent attains this with high efficiency: the planner requires only **one demonstration per task**, and the executor is trained on a single server equipped with $8\times$ **NVIDIA A100 80GB GPUs**. Moreover, K²-Agent exhibits dual generalization that supports our core hypothesis: (1) Declarative knowledge transfer—the high-level planner's learned knowledge transfers across backbone models; and (2) Procedural knowledge transfer—the executor's learned skills generalize to entirely novel tasks on Android-in-the-Wild (AitW) (Rawles et al., 2023) and ScreenSpot-v2 (Wu et al., 2024).

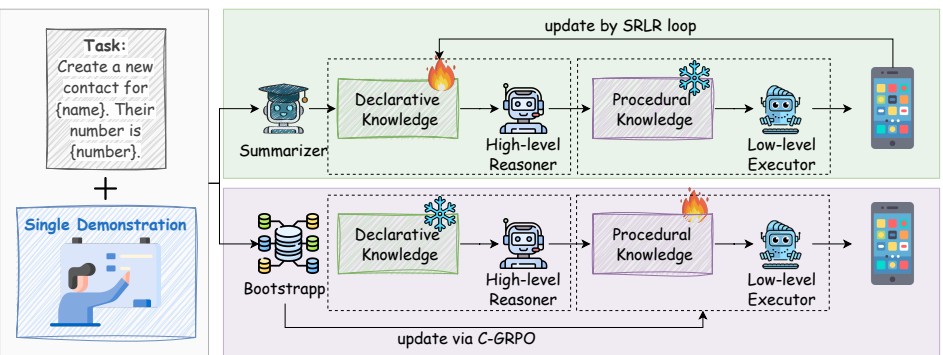

Figure 2: **Overview of the K²-Agent**. **Top**: The SRLR loop where declarative knowledge (knowing what) is iteratively improved using feedback. **Bottom**: The skill acquisition process where procedural knowledge (knowing how) is learned via C-GRPO, bootstrapped from a single demonstration.

To summarize, our contributions are as follows:

- We introduce K²-Agent, a hierarchical framework that enables closed-loop co-evolution of declarative and procedural knowledge.

- We design a single-demonstration–initiated SRLR cycle that continuously incorporates execution feedback to refine a task knowledge base.

- We propose a curriculum-guided RL algorithm (C-GRPO) that efficiently acquires procedural skills via error-decoupled experience-pool balancing and dynamic demonstration injection.

- We provide extensive evidence of dual generalization—declarative knowledge across backbones and procedural skills across benchmarks—highlighting the critical role of co-evolution in enhancing generalization.

## 2 RELATED WORK

### 2.1 TRAINING-FREE AGENTS

Training-free agents leverage the in-context learning capabilities of large vision-language models (VLMs) to solve mobile control tasks via carefully engineered prompts and reasoning loops (Zhang et al., 2025; Wang et al., 2024b). Recent works have explored various mechanisms, such as building explicit knowledge bases (Zhang et al., 2025), incorporating reflection steps (Wang et al., 2024b), or adopting multi-agent (Wang et al., 2024a) and hierarchical architectures (Agashe et al., 2024; 2025; Wang et al., 2025b). While these methods excel at leveraging the fixed knowledge of powerful foundation models, their self-improvement mechanisms are typically non-parametric (e.g., memory editing). In contrast, K²-Agent introduces a hybrid approach: it uses a non-parametric SRLR loop to evolve declarative knowledge while simultaneously fine-tuning a dedicated executor with C-GRPO to parametrically improve its procedural skills through interaction.

### 2.2 LEARNING-BASED AGENT

Learning-based agents fine-tune models on domain-specific data to adapt them for mobile control. The field has rapidly progressed from initial SFT on GUI understanding (Hong et al., 2024; Zhang & Zhang, 2023) to RL for interactive decision-making (Bai et al., 2024; Wang et al., 2024c). With the advent of advanced policy optimization techniques like DPO (Rafailov et al., 2023) and GRPO (Shao et al., 2024), recent models have achieved significant gains in grounding and task success by post-training strong open-source VLMs (Qin et al., 2025; Lu et al., 2025; Luo et al., 2025; Liu et al., 2025b; Gu et al., 2025). However, these methods typically train a single, monolithic policy, which conflates the learning of high-level task strategy ("knowing what") and low-level action execution ("knowing how"). K²-Agent's core distinction is the explicit decoupling of these two learning processes. By using different, specialized update rules for declarative and procedural knowledge, our framework enables more targeted, data-efficient, and effective learning for both.

## 3 PRELIMINARIES

We model mobile device control as a finite-horizon Markov Decision Process (MDP) $M = (\mathcal{S}, \mathcal{A}, \mathcal{P}, \mathcal{R}, \gamma)$. A state $s_t \in \mathcal{S}$ is a multimodal representation $s_t = (o_t, g)$, where $o_t$ is the current visual observation of the screen (we use **only raw screenshots**, without any extra UI metadata such as the accessibility tree) and $g$ is the task instruction. The action space $\mathcal{A}$ consists of parameterized primitive UI operations, such as `click`, `long-press`, `swipe`, and `type`. The transition function $\mathcal{P}(s_{t+1} \mid s_t, a_t)$ defines the probability of moving to $s_{t+1}$ after taking $a_t$ in $s_t$. The reward function $\mathcal{R}$ provides environmental feedback. This feedback can be sparse trajectory-level rewards, indicating overall task success; or dense step-level rewards, guiding fine-grained action generation. The discount factor is $\gamma$. The agent's objective is to learn a policy $\pi(a_t \mid s_t)$ that maximizes the expected return: $J(\pi) = \mathbb{E}_{\tau \sim \pi} \left[ \sum_{t=0}^{T} \gamma^t r_t \right]$.

## 4 METHOD

We propose **K²-Agent**, a hierarchical framework that mirrors human cognition by separating and co-evolving declarative ("knowing what") and procedural ("knowing how") knowledge. Section 4.1 outlines our design. Section 4.2 details the high-level planner that evolves declarative knowledge through a SRLR self-improvement loop. Section 4.3 introduces C-GRPO for learning procedural skills in the low-level executor. Section 4.4 provides implementation and training details.

### 4.1 OVERVIEW OF THE K²-AGENT FRAMEWORK

As shown in Figure 2, **K²-Agent** features a two-layer Planner-Executor architecture, where each layer is initialized by a VLM. The high-level planner, $\pi_H$, operates in a training-free mode to maintain a declarative knowledge base, $K_G$, which is iteratively refined via our SRLR loop. Rather than acting directly on the environment, $\pi_H$ consults $K_G$ to decompose the global task $g$ into a sequence of immediate sub-goals, $z_t$. The low-level executor, $\pi_L$, is a trainable policy that acquires procedural skills via C-GRPO algorithm. It receives the current observation $o_t$ and the sub-goal $z_t$ from $\pi_H$, making decisions in an augmented state $s'_t = (o_t, g, z_t)$ to produce atomic actions on the device.

The two modules form a closed-loop *co-evolution* system. Forward communication occurs via the sub-goals $z_t$. The feedback loop consists of execution outcomes—successes, failures, and error patterns—from $\pi_L$ being used by $\pi_H$ to revise the knowledge base $K_G$. A more accurate $K_G$ allows $\pi_H$ to generate more feasible and executable sub-goals, in turn providing $\pi_L$ with a more structured exploration problem and thus more effective learning signals. This creates a synergistic cycle where improved planning and execution reinforce one another.

### 4.2 HIGH-LEVEL PLANNER: EVOLVING DECLARATIVE KNOWLEDGE VIA SRLR LOOP

The planner, $\pi_H$, evolves its declarative knowledge base, $K_G$, through a four-stage SRLR loop, illustrated in Figure 3. This self-improvement cycle is initiated by a single expert demonstration and performed by the VLM-based planner itself. We detail each stage below. Complete implementation details and a case study of $K_G$ evolution are provided in Appendix B.1.

#### 4.2.1 SUMMARIZE

Given a single demonstration trajectory $\mathcal{T}^d = \{(s_t^d, a_t^d)\}_{t=0}^{T^d}$ and a task goal $g$, $\pi_H$ performs a one-pass distillation to produce a structured initial task knowledge base $K_G^0$ (Figure 3, top-left):

$$K_G^0 = \text{Summarize}(\mathcal{T}^d, g; \theta_H). \tag{1}$$

The knowledge base is represented as a set of rules or a stepwise checklist that captures the core logic for completing the task, key UI elements, and their functions. It serves as the starting point for subsequent iterations in the SRLR loop.

#### 4.2.2 REFLECT

During execution, the reflection module is activated upon completing a new trajectory $\mathcal{T}^e$. Reflection operates at two granularities:

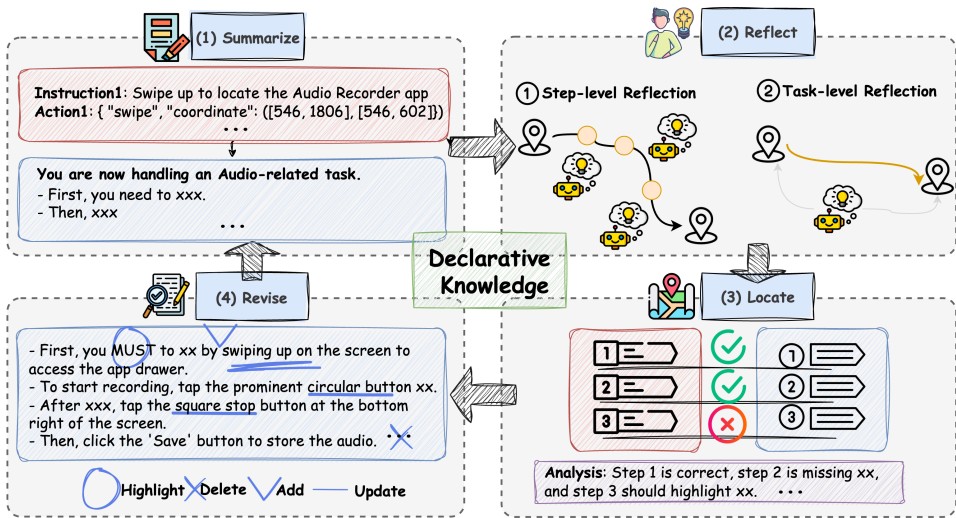

Figure 3: An illustration of the SRLR self-evolution loop. (1) Summarize: An initial knowledge base is automatically distilled from a demonstration. (2) Reflect: The agent analyzes its execution trace to identify deviations. (3) Locate: The failure's root cause is pinpointed. (4) Revise: Atomic operators repair the knowledge base for the next iteration.

**Step-level.** The planner continuously verifies whether each action's outcome aligns with the expected result in $K_G$. This allows for the immediate detection of deviations from the plan.

**Task-level.** If the episode fails, task-level reflection analyzes the entire trajectory to generate a structured, root-cause explanation for the failure, such as "failed to identify the `Rename` button":

$$M^{\text{case}} = \text{Reflect}_{\text{task}}(\mathcal{T}^e, K_G, g; \theta_H).\qquad(2)$$

### 4.2.3 LOCATE

To enable precise revision, the *locate* module aligns the executed trajectory $\mathcal{T}^e$ with the task knowledge encoded in $K_G$ and identifies the first decision point that yields an unexpected outcome:

$$t^* = \text{Locate}(\mathcal{T}^e, K_G; \theta_H) = \min\Big\{ t \ \Big| \ \text{Verify}\big(s_{t+1}^e, a_t^e, K_G, t; \theta_H\big) = \text{False}\Big\}.\qquad(3)$$

Here, $\text{Verify}(\cdot) \in \{\text{True, False}\}$ checks whether executing $a_t^e$ in state $s_t^e$ produces a next state $s_{t+1}^e$ that matches the expected outcome for critical step $t$ specified by $K_G$.

### 4.2.4 REVISE

Finally, given the failure explanation $M^{\text{case}}$ and the failure point $\big(t^*, s_{t^*}^e\big)$, the system performs local surgeries on $K_G$ using four atomic operators: **Add** inserts missing steps; **Delete** removes erroneous instructions; **Update** modifies parameters; and **Highlight** emphasizes critical constraints. These operations yield a revised version $K_G'$ (see evolution example in Appendix B.1):

$$K_G' = \text{Revise}\big(K_G, \ \big(t^*, s_{t^*}^e\big), \ M^{\text{case}}; \ \theta_H\big).\qquad(4)$$

By iterating the SRLR loop, the $K_G$ improve over time, enabling higher-quality planning.

### 4.3 LOW-LEVEL EXECUTOR: LEARNING PROCEDURAL SKILLS WITH C-GRPO

Training the low-level executor, $\pi_L$, faces two challenges: (i) Sample Imbalance: The training data is often biased, with common operations (e.g., `click`) heavily more than rare ones (e.g., `long-press`, `swipe`). (ii) Inefficient Exploration: In long-horizon tasks, the huge action space makes it difficult for agents to autonomously discover successful trajectories through trial and error.

To address these issues, we propose Curriculum-Guided Group Relative Policy Optimization (C-GRPO), an algorithm that efficiently acquires procedural skills via a novel error-decoupled replay balancing mechanism and a dynamic demonstration injection strategy.

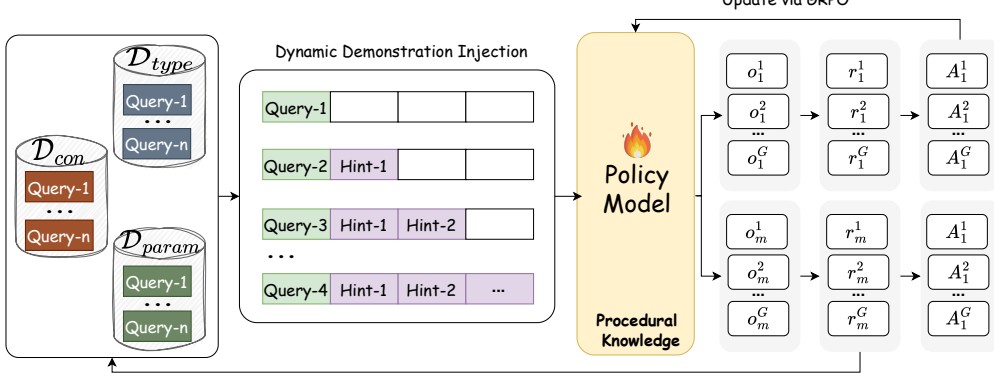

Figure 4: Our C-GRPO framework, featuring its two main curriculum components: Error-Decoupled Replay Balancing (left) to construct balanced mini-batches, and Dynamic Demonstration Injection (center) to provide adaptive guidance for the GRPO update (right).

### 4.3.1 ERROR-DECOUPLED REPLAY BALANCING

We observe that execution errors at the action level can be decoupled into *type errors* (e.g., predicting `swipe` instead of `click`) and *parameter errors* (e.g., a `click` with inaccurate coordinates). For any training input $i$, the low-level model $\pi_L$ generates $G$ candidate actions $\{a_i^{(g)}\}_{g=1}^G$. Given an expert action $\hat{a}_i$, we define a binary reward

$$r(a, \hat{a}) = \mathbb{1}\Big[\text{type}(a) = \text{type}(\hat{a}) \ \wedge \ \big\|\text{coord}(a) - \text{coord}(\hat{a})\big\|_2 < \epsilon\Big], \tag{5}$$

where $\text{type}(\cdot)$ denotes the primitive operator (`click`, `long-press`, `swipe`, `type`), and $\text{coord}(\cdot) \in \mathbb{R}^2$ denotes the action's spatial parameters.

Using the $G$ candidates, we estimate two error rates per input $i$:

$$\eta_{\text{type}}(i) = \frac{1}{G} \sum_{g=1}^G \mathbb{1}\big[\text{type}(a_i^{(g)}) \neq \text{type}(\hat{a}_i)\big], \tag{6}$$

$$\eta_{\text{param}}(i) = \frac{1}{G} \sum_{g=1}^G \mathbb{1}\Big[\text{type}(a_i^{(g)}) = \text{type}(\hat{a}_i) \ \wedge \ \big\|\text{coord}(a_i^{(g)}) - \text{coord}(\hat{a}_i)\big\|_2 \geq \epsilon\Big]. \tag{7}$$

Based on $\eta_{\text{type}}(i)$ and $\eta_{\text{param}}(i)$, each input is dynamically assigned to one of three replay buffers: the *conventional pool* $\mathcal{D}_{\text{con}}$, the *type-exploration pool* $\mathcal{D}_{\text{type}}$, and the *precision-optimization pool* $\mathcal{D}_{\text{param}}$. During training, each mini-batch is formed by sampling from these buffers with preset ratios $\{\beta_{\text{con}}, \beta_{\text{type}}, \beta_{\text{param}}\}$, ensuring balanced progress on the model's different weaknesses, leading to more efficient overall improvement.

### 4.3.2 DYNAMIC DEMONSTRATION INJECTION

For mobile control agents built on (V)LLMs, the action space effectively spans a vast textual space grounded to the entire screen. Replay balancing alone cannot make the model reliably discover the correct action sequence in such a large space, so rewards remain sparse on complex tasks. To guide exploration, we introduce a dynamic demonstration injection mechanism that prepends a variable number of atomic expert actions to the input. The injected length $l$ (in steps) is scheduled by

$$l = L_h(k, d_i) = L \cdot \sigma(k) \cdot f_{\text{gate}}(d_i), \tag{8}$$

where $L$ is the total number of steps in the full demonstration; $\sigma(k) = \max(0, 1 - k/K_{\max})$ is a linear annealing scheduler that decays with the training step $k$; $f_{\text{gate}}(d_i) = \tanh(d_i/T)$ is a difficulty-gating function controlled by a temperature $T$; and the sample difficulty score $d_i$ is the total error rate, $d_i = \eta_{\text{type}}(i) + \eta_{\text{param}}(i)$. The actual prefix consists of the first $\lfloor l \rfloor$ complete actions.

Intuitively, this mechanism provides more extensive guidance for samples the model currently finds difficult, while gradually weaning the model off all demonstrations as training progresses. This curriculum-based approach greatly increases the probability of generating successful trajectories, thereby producing denser and higher-quality signals for policy optimization.

### 4.3.3 C-GRPO OBJECTIVE

Our final objective integrates these curriculum strategies with the GRPO framework. In each training step, we construct a balanced mini-batch $\mathcal{B}$ and, for each sample $q \in \mathcal{B}$, form an augmented context $c$ using dynamic demonstration injection. The policy is then updated by maximizing the GRPO objective, where the advantage estimates $\hat{A}_{i,t}$ are derived from group-relative rewards based on the dense, binary expert-matching signal defined in Equation 5. The objective is:

$$\mathcal{J}_{\text{C-GRPO}}(\theta; c) = \mathbb{E}_{\{o_i\}_{i=1}^G \sim \pi_{\theta_{\text{old}}}(\cdot|c)} \left[ \frac{1}{G} \sum_{i=1}^{G} \frac{1}{|o_i|} \sum_{t=1}^{|o_i|} \min\left( r_{i,t}(\theta) \hat{A}_{i,t}, \text{ clip}\left(r_{i,t}(\theta), 1-\epsilon, 1+\epsilon\right) \hat{A}_{i,t} \right) \right],$$

(9)

where the importance ratio $r_{i,t}(\theta) = \frac{\pi_\theta(o_{i,t}|c,o_{i,<t})}{\pi_{\theta_{\text{old}}}(o_{i,t}|c,o_{i,<t})}$. By optimizing this objective, C-GRPO learns robust procedural skills with denser and more informative signals.

### 4.4 TRAINING AND IMPLEMENTATION DETAILS

We use Qwen-2.5-VL-72B as the high-level planner ($\pi_H$) in the training-free SRLR loop, and the parameter-efficient Qwen-2.5-VL-7B as the low-level executor ($\pi_L$), which is post-trained with C-GRPO. The planner's knowledge base ($K_G$) is bootstrapped from a single expert demonstration, which we recorded for each task type through a dedicated data collection pipeline (data collection, cleaning, and scale are reported in Appendix B.1.2). For the learning-based executor, the C-GRPO training set is constructed from demonstration samples collected across all AndroidWorld tasks, ensuring a strict separation between training and test sets. The executor's training is guided by a dense, step-level reward signal composed of a binary format reward (for syntactic correctness) and a content reward (Equation 5, for semantic correctness). Further details on data format, dataset scale, and reward computation are provided in Appendix B.2.

To conserve computational resources, the co-evolution of the planner and executor is implemented via an efficient alternating update mechanism that follows the pattern $(\text{SRLR}_H)^n \to \text{C-GRPO}_L$. We set $n = 3$ in our experiments, where the planner first refines its knowledge base $K_G$ over several iterations before the executor undergoes a single, intensive training phase. All training was conducted on a single server equipped with $8\times$ NVIDIA A100 80GB GPUs. A complete list of hyperparameters used during training is available in Appendix B.2.3.

## 5 EXPERIMENTAL EVALUATION

We conduct extensive experiments to evaluate our $\mathbf{K^2}$-**Agent**. This section is organized as follows. In Section 5.1, we evaluate on the AndroidWorld benchmark and achieve a new state-of-the-art (SOTA). In Section 5.2, we study generalization along two axes: transfer of high-level declarative knowledge across different backbones, and transfer of low-level procedural skills across benchmarks. In Section 5.3, we present ablations that quantify the contribution of each core component.

### 5.1 PERFORMANCE AGAINST BASELINES

**Environment.** We evaluate on AndroidWorld(Rawles et al., 2024), a widely used benchmark with 116 tasks across 20 apps. Each task is instantiated with randomized parameters per episode, preventing overlap between the training and test splits. Tasks are categorized by difficulty (easy/medium/hard). Human experts achieve about $80\%$ (Rawles et al., 2024) average success on this platform.

**Baselines.** We compare $K^2$-Agent to two families of methods: (1) *Training-free*: agents that rely on few-shot capabilities of (V)LLMs (e.g., GPT series (Achiam et al., 2023), Claude-3.7 Sonnet (Anthropic, 2025a), Gemini 2.5 Pro (Comanici et al., 2025)). (2) *Learning-based*: agents that fine-tune open-source backbones with domain data via SFT and/or RL.

Table 1: Comparison of $K^2$-Agent and baselines on AndroidWorld. Our method reports the mean ± std over 3 independent runs.

| Type | Agent | Base Model | Input | SR |
|---|---|---|---|---|
| *Training free* | M3A (Rawles et al., 2024) | GPT-4 Turbo | Screenshot + A11y | 30.6 |
| | AndroidGen (Lai et al., 2025) | GPT-o | A11y Tree | 46.8 |
| | Agent S2 (Agashe et al., 2025) | Claude-3.5-Sonnet | Screenshot | 54.3 |
| | MobileUse (Li et al., 2025) | Qwen2.5-VL-72B | Screenshot | 62.9 |
| | DroidRun (DroidRun, 2025) | Gemini-2.5-Pro | Screenshot + A11y | 63.0 |
| | FinalRun (FinalRun, 2025) | GPT-5 | Screenshot + A11y | 76.7 |
| *Learning based* | InfiGUI Agent (Liu et al., 2025a) | Qwen2-VL-2B | Screenshot | 9.0 |
| | GUI-critic-R1 (Wanyan et al., 2025) | Qwen2.5-VL-2B | Screenshot + A11y | 27.6 |
| | UI-TARS (Qin et al., 2025) | Qwen-2-VL-72B | Screenshot | 46.6 |
| | UI-Venus (Gu et al., 2025) | Qwen2.5-VL-72B | Screenshot | 65.9 |
| | Seed1.5-VL (Guo et al., 2025) | Seed1.5-VL-72B | Screenshot + A11y | 62.1 |
| | Mobile-Agent-v3 (Ye et al., 2025) | Qwen-VL based | Screenshot | 73.3 |
| | UI-TARS-2 (Wang et al., 2025a) | Seed-thinking-1.6 | Screenshot | 73.3 |
| | AutoGLM-Mobile (Liu et al., 2024) | AutoGLM-Mobile | Screenshot + A11y | 75.8 |
| *Ours* | **$K^2$-Agent** | Qwen2.5-VL (72B+7B) | Screenshot | **76.1 ± 1.0** |

**Results and analysis.** Table 1 reports success rates on AndroidWorld. $K^2$-Agent sets a new **SOTA** with a **76.1%** average success rate, surpassing the strongest open-source learning-based methods, `UI-TARS-v2` (73.3%) and `Mobile-Agent-v3` (73.3%), and outperforming all closed-source models restricted to screenshot inputs. Beyond accuracy, our method offers two key advantages: (i) **Screenshot-only input**—unlike many high-ranked methods that exploit accessibility (A11y) trees, $K^2$-Agent operates solely from raw screenshots; (ii) **Optimization Efficiency**—the high-level model is bootstrapped from a single demonstration per task category, while the low-level executor builds on a 7B open-source backbone, requiring substantially fewer training resources.

## 5.2 MODEL GENERALIZATION

$K^2$-Agent enables two forms of generalization by our design: (i) high-level *declarative knowledge* transfers across planner backbones, and (ii) low-level *procedural skills* transfer across benchmarks.

**Transfer of declarative knowledge.** The SRLR-produced knowledge $K_G$ is language-based, and explicit. We reuse the same $K_G$ (**without extra tuning**) across different VLMs and re-evaluate on AndroidWorld. Figure 5 (a) reports task success rates. All backbones benefit from the injected $K_G$, indicating that the distilled declarative knowledge is model-agnostic and broadly reusable.

**Transfer of procedural skills.** We directly transfer the low-level executor, trained on Android-World, to the ScreenSpot-v2 benchmark in a zero-shot setting. As shown in Table 3, $K^2$-Agent's executor achieves a **91.3%** overall accuracy, outperforming general-purpose closed-source models like Claude 3.7 Sonnet and reaching a level competitive with specialized agents such as GUI-Owl-32B, which were trained on massive-scale GUI datasets. We further validate transfer to **Android-in-the-Wild (AitW)** across two subsets, where the high-level planner is bootstrapped from a single demonstration per subset and the low-level executor is transferred directly. $K^2$-Agent surpasses existing RL- and SFT-based approaches as well as closed-source models. Results are shown in Table 2 and Table 3. Additional analysis is provided in Appendix E.1.2.

Table 2: Zero-shot transfer to AitW using the low-level executor trained on AndroidWorld.

| Agent | AitW-General (SR %) | AitW-WebShopping (SR %) |
|---|---|---|
| SoM (Zheng et al., 2024) | 16.7 | 11.5 |
| AppAgent (Zhang et al., 2025) | 17.7 | 8.3 |
| CogAgent (Hong et al., 2024) | 25.0 | 38.5 |
| AutoUI (Zhang & Zhang, 2023) | 22.9 | 25.0 |
| DigiRL (Bai et al., 2024) | 71.9 | 67.2 |
| **$K^2$-Agent** | **86.5** | **68.3** |

Table 3: Zero-shot performance of the K²-Agent executor on the ScreenSpot-v2 benchmark.

| Agent Model | Mobile | | Desktop | | Web | | Overall |
|---|---|---|---|---|---|---|---|
| | Text | Icon | Text | Icon | Text | Icon | |
| Operator (OpenAI, 2025) | 47.3 | 41.5 | 90.2 | 80.3 | 92.8 | 84.3 | 70.5 |
| Claude 3.7 Sonnet (Anthropic, 2025b) | - | - | - | - | - | - | 87.6 |
| UI-TARS-72B (Qin et al., 2025) | 94.8 | 86.3 | 91.2 | 87.9 | 91.5 | 87.7 | 90.3 |
| JEDI-7B (Xie et al., 2025) | 96.9 | 87.2 | 95.9 | 87.9 | 94.4 | 84.2 | 91.7 |
| GUI-Owl-32B (Ye et al., 2025) | 98.6 | 90.0 | 97.9 | 87.8 | 94.4 | 86.7 | 93.2 |
| UI-Venus-Ground-72B (Gu et al., 2025) | 99.7 | 93.8 | 95.9 | 90.0 | 96.2 | 92.6 | **95.3** |
| K²-Agent | 96.9 | 80.6 | 95.9 | 83.6 | 95.3 | 90.6 | 91.3 |

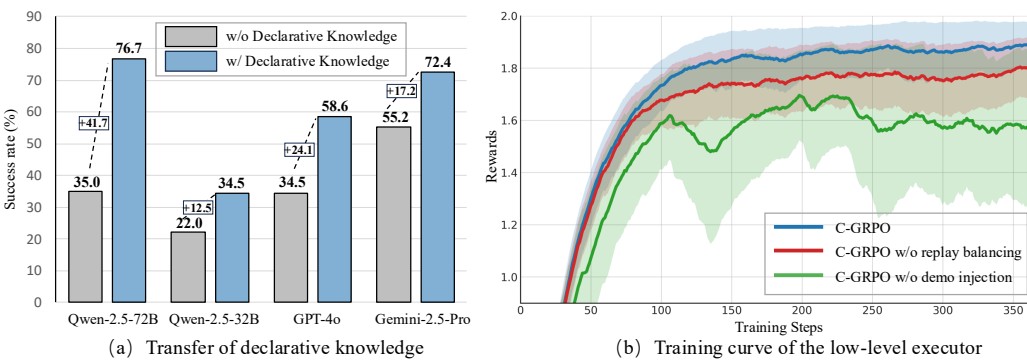

(a) Transfer of declarative knowledge  (b) Training curve of the low-level executor

Figure 5: **Ablation and Component Analysis. (a)**: The declarative knowledge from our SRLR loop provides a substantial performance boost across four different powerful VLM backbones. **(b)**: Training reward curves for the low-level executor comparing full C-GRPO (blue), C-GRPO without replay balancing (red), and C-GRPO without demonstration injection (green).

## 5.3 ABLATION STUDIES

To assess the contribution of each component in K²-Agent, we conduct ablations on AndroidWorld by removing or replacing specific modules. We compare five configurations: **(1) No Hierarchy.** A flat, end-to-end model. **(2) No Hierarchy + SRLR**: A flat model that directly incorporates the knowledge base $K_G$ from the SRLR loop. **(3) Hierarchical (SRLR + SFT-Low).** Our hierarchical design with an SRLR planner, while the low-level executor is trained only with SFT. **(4) Hierarchical (SRLR + GRPO-Low).** The hierarchical design with a vanilla GRPO-trained executor. **(5) K²-Agent (Full).** Our complete model with the SRLR planner and C-GRPO executor.

Results are summarized in Table 4 and Figure 1. The *No Hierarchy* model performs poorly, confirming that a flat architecture struggles to manage both planning and execution. Simply adding the SRLR knowledge base (*No Hierarchy + SRLR*) provides a notable boost, demonstrating the value of explicit declarative knowledge. A significant leap occurs when we introduce the hierarchy (*SRLR + SFT-Low*), isolating the structural benefit of decoupling know-what from know-how. Within the hierarchy, replacing SFT with vanilla GRPO improves performance further by enabling interactive learning, but progress is limited by inefficient exploration. Finally, our full **K²-Agent** with C-GRPO achieves the highest success rate. The superiority of C-GRPO over vanilla GRPO

Table 4: Ablation study of K²-Agent components on Android-World benchmark.

| Configuration | SR (%) |
|---|---|
| No Hierarchy | 35.3 |
| No Hierarchy + SRLR | 58.6 |
| SRLR + SFT-Low | 62.0 |
| SRLR + GRPO-Low | 68.9 |
| **K²-Agent (Full)** | 76.1 |

is not only reflected in the final success rate but also evident during training, as shown in Figure 5 (b), where C-GRPO consistently achieves higher and more stable rewards.

We further dissect the training dynamics of the low-level executor by isolating the contribution of the two key components in C-GRPO. As shown in Figure 5(b), the full C-GRPO framework achieves the highest rewards and the most stable convergence. Ablating *Dynamic Demonstration Injection* (green curve) leads to the most severe degradation: the policy attains substantially lower rewards with pro-

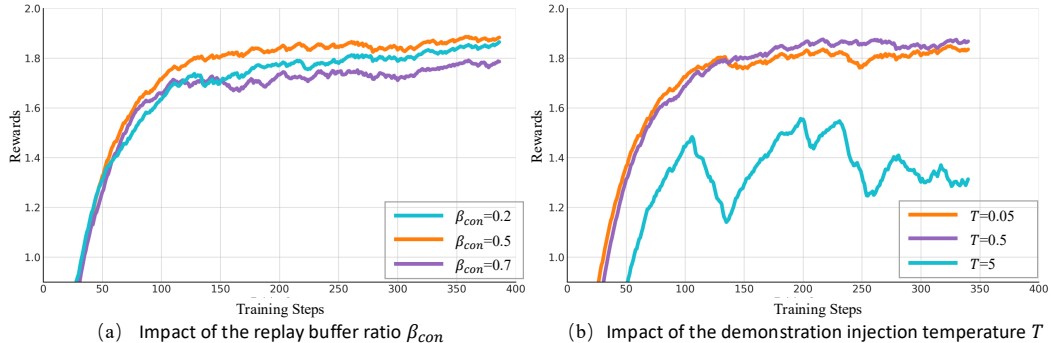

Figure 6: **Parameter sensitivity of C-GRPO. (a)**: Varying the conventional pool ratio $\beta_{\mathrm{con}}$. **(b)**: Effect of the injection temperature $T$ in the difficulty-gating function.

nounced oscillations throughout training. This indicates that expert-prefixed trajectories are crucial for bootstrapping exploration, enabling the agent to reliably discover successful behaviors early and thereby stabilizing subsequent self-generated rollouts. Meanwhile, removing *Error-Decoupled Replay Balancing* (red curve) results in a visibly slower convergence rate and slightly inferior final performance, suggesting that without balancing experience across error types, the optimizer is biased towards frequent, easier operations and struggles to acquire complex skills.

## 5.4 PARAMETER SENSITIVITY

We analyze the sensitivity of C-GRPO to two key hyperparameters: the conventional replay pool ratio $\beta_{\mathrm{con}}$ and the demonstration injection temperature $T$. Full grid-search settings for all hyperparameters are reported in Appendix B.2.3.

**Impact of Replay Balancing Ratios.** The left panel of Figure 6 shows training dynamics under different $\beta_{\mathrm{con}}$, with the remaining probability split equally between $\mathcal{D}_{\mathrm{type}}$ and $\mathcal{D}_{\mathrm{param}}$. All settings exhibit a clear upward trend and eventually converge, indicating that C-GRPO is robust to replay buffer composition. Among them, $\beta_{\mathrm{con}} = 0.5$ strikes the best balance between preserving the natural rollout distribution and emphasizing error-corrective samples, yielding the fastest convergence and highest final reward. In contrast, assigning excessive weight to the conventional pool (e.g., $\beta_{\mathrm{con}} = 0.7$) reduces exposure to hard examples and slightly slows learning.

**Impact of Injection Temperature.** The right panel of Figure 6 examines the injection temperature $T$, which controls the difficulty-gating function $f_{\mathrm{gate}}(d_i) = \tanh(d_i/T)$. With a large temperature ($T = 5$), $f_{\mathrm{gate}} \approx 0$, leading to very short or absent expert prefixes and effectively reverting C-GRPO to vanilla GRPO; the policy barely improves and fails to converge. In contrast, lower temperatures ($T = 0.05, 0.5$) activate the curriculum effectively: $T = 0.05$ yields a slightly faster initial improvement due to more aggressive intervention, while $T = 0.5$ provides smoother training and slightly better stability and final performance. Overall, C-GRPO remains stable over a reasonable range of hyperparameters, and we adopt $\beta_{\mathrm{con}} = 0.5$ and $T = 0.5$ in all main experiments.

## 6 CONCLUSION

We propose K²-Agent, a hierarchical framework inspired by the cognitive separation of declarative ("knowing what") and procedural ("knowing how") knowledge. Our agent synergistically co-evolves these two capabilities distinctly: a high-level planner uses an SRLR loop to distill and self-evolving task knowledge from a single demonstration, while a low-level executor masters precise actions via our highly-efficient C-GRPO post-training algorithm. K²-Agent not only achieves SOTA performance on AndroidWorld but, more critically, exhibits powerful and robust dual generalization—transferring declarative knowledge across backbones and procedural skills across benchmarks. We believe that this knowledge decoupling and co-evolution framework offers a promising new paradigm for building more general, efficient, and adaptable agents.

ACKNOWLEDGEMENT

This work is supported by the National Key Research and Development Plan under Grant No. 2024YFB4505500 & 2024YFB4505503.

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

APPENDIX

**Statement on LLM Usage.** In the preparation of this manuscript, Large Language Models (LLMs) were utilized as auxiliary tools to enhance language quality and formatting. Specifically, we used Gemini 2.5 Pro for grammar checking and polishing the prose. Additionally, GPT-series models were employed to assist with optimizing LaTeX table formatting and to query for specific typesetting commands (e.g., for pseudocode presentation and color highlighting). We affirm that all scientific content, core ideas, and experimental results were conceived and articulated entirely by the human authors. LLMs were not used to generate any substantive scientific content. All suggestions and modifications from these models were implemented under the direct supervision and final approval of the authors. All co–authors are aware of and consent to this usage.

## A   COGNITIVE SCIENCE FOUNDATIONS OF K²-AGENT

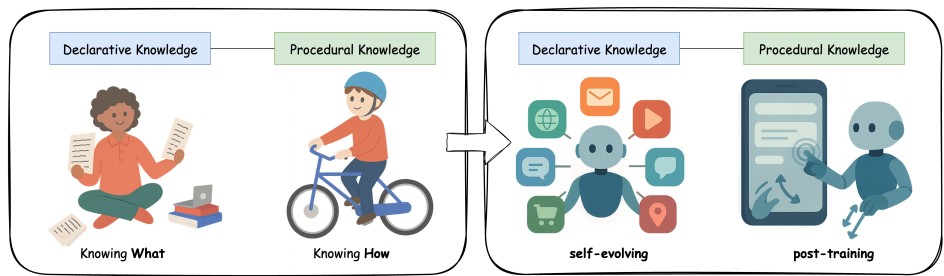

Figure 7: **Cognitive inspiration for the K²-Agent framework.** The design maps the human distinction between declarative ("knowing what") and procedural ("knowing how") knowledge onto a hierarchical agent architecture.

Figure 7 illustrates the core cognitive science principle that inspires the design of K²-Agent: the fundamental distinction between declarative ("knowing what") and procedural ("knowing how") knowledge. Seminal work in cognitive neuroscience (Squire & Knowlton, 1995) has established these as distinct memory systems. Declarative knowledge consists of explicit facts and concepts that can be consciously recalled and articulated, much like how our high-level planner distills and refines its task knowledge base ($K_G$) from a single demonstration. In contrast, procedural knowledge encompasses implicit skills acquired through repeated practice, such as riding a bicycle, which are performed automatically and are difficult to verbalize. This mirrors how our low-level executor is trained via C-GRPO to form robust "muscle memory" for precise UI operations. By explicitly modeling this cognitive division, K²-Agent creates a synergistic architecture where planning and execution can be evolved and optimized using distinct, more suitable mechanisms.

## B   FRAMEWORK AND ALGORITHM DETAILS

### B.1   HIGH-LEVEL PLANNER: THE SRLR LOOP IMPLEMENTATION

#### B.1.1   PSEUDOCODE FOR THE SRLR ITERATION

The iterative self-evolution of the high-level planner is governed by the Summarize–Reflect–Locate–Revise (SRLR) loop. Algorithm 1 provides a detailed algorithmic view of this process, outlining how the knowledge base ($K_G$) is initialized from a single demonstration and then progressively refined through cycles of execution and feedback-driven revision.

The SRLR loop is guided by two primary hyperparameters that control its termination. **'MAX_ITER'** sets a hard limit on the number of revision cycles to prevent infinite loops, which we set to 10 in our experiments. **'SUCCESS_THRESH'** defines a stopping criterion based on consistent performance; the loop terminates if the agent successfully completes the task for this many consecutive episodes. We set this value to 3, indicating that the knowledge base is considered stable and robust after three successful runs in a row.

---

**Algorithm 1:** The SRLR Algorithm for Planner Self-Evolution

---

**Input:** Demonstration $\mathcal{T}^d$, task goal $g$, environment $\mathcal{E}$, VLM backbone $\theta_H$
**Initialize:** Evolved knowledge base $K_G$

// Summarize: Distill initial knowledge from a single demonstration
1   $K_G \leftarrow \text{Summarize}(\mathcal{T}^d, g; \theta_H)$

// Iterative Refinement: The main SRLR loop
2   $i \leftarrow 0, s_{\text{streak}} \leftarrow 0$
3   **while** $i < MAX\_ITER$ **and** $s_{streak} < SUCCESS\_THRESH$ **do**
4      $i \leftarrow i + 1$
5      $(\mathcal{T}^e, \text{success}) \leftarrow \text{Execute}(\pi_H, \pi_L, g, K_G, \mathcal{E})$          // Execute with current $K_G$
6      **if** *success* **then**
7         $s_{\text{streak}} \leftarrow s_{\text{streak}} + 1$
8      **else**
9         $s_{\text{streak}} \leftarrow 0$
        // Reflect: Analyze the root cause of the failure
10        $M^{\text{case}} \leftarrow \text{Reflect}(\mathcal{T}^e, K_G, g; \theta_H)$
        // Locate: Pinpoint the first point of failure
11        $t^* \leftarrow \text{null}$
12        **for** $t = 0$ **to** $|\mathcal{T}^e| - 1$ **do**
13           **if** $\text{Verify}(s_{t+1}^e, a_t^e, K_G, t; \theta_H) = \text{False}$ **then**
14             $t^* \leftarrow t$; **break**
15        **if** $t^* = null$ **then**
16          $t^* \leftarrow \text{FullTrajectoryAnalysis}(\mathcal{T}^e, \mathcal{T}^d, K_G; \theta_H)$      // Fallback if loop fails
        // Revise: Intelligently update the knowledge base
17        **if** $t^* \neq null$ **then**
18          $\Delta K_G \leftarrow \text{GenerateRevision}(M^{\text{case}}, (\mathcal{T}^e, t^*), \mathcal{T}^d; \theta_H)$
19          $K_G^{\text{draft}} \leftarrow \text{IntelligentFusion}(K_G, \Delta K_G; \theta_H)$
20          $K_G \leftarrow \text{ApplyAtomicEdits}(K_G^{\text{draft}}, \Delta K_G)$

21   **return** $K_G$

---

### B.1.2   DEMONSTRATION DATA CONSTRUCTION

To initialize the SRLR self-evolution loop for the high-level planner $\pi_H$, we constructed a high-quality human expert demonstration trajectory, $\mathcal{T}^d$, for each task category. This demonstration data was recorded by a human operator in a standard Android emulator environment. For each step, the operator was instructed to first articulate their intent as a natural language instruction (e.g., "Swipe up the screen to locate the Audio Recorder app.") while the system precisely recorded the corresponding atomic action, including its type and exact coordinate parameters. We also captured screenshots immediately before and after each action was executed. A complete demonstration trajectory thus consists of a sequence of steps, where each step comprises a tuple: (pre-operation screenshot, post-operation screenshot, natural language instruction, atomic action). For example, in the "ContactsAddContact" task, the trajectory contains 10 steps; the natural language instruction for the first step is "Swipe up on the screen to locate the Contacts app.", and its corresponding atomic action is `[swipe, (546, 1806), (546, 800)]`.

For the 116 unique task categories in AndroidWorld, we collected a total of **103** expert demonstration trajectories to bootstrap the SRLR process. This number is less than the total task count for two primary reasons. (i) some tasks are inherently difficult even for human experts to complete reliably, and (ii) certain tasks share overlapping high-level knowledge, allowing one trajectory to effectively cover multiple categories. In principle, this set could be further reduced by exploiting such knowledge sharing across tasks.

We maintained a strict separation between the demonstration data used to bootstrap the SRLR loop and the test data used for evaluation. To ensure this, we followed the design of the AndroidWorld

benchmark by setting different random seeds for each task instance to dynamically generate its parameters. This mechanism guarantees that even for tasks of the same category, the specific parameters encountered during testing (e.g., contact names, filenames, or settings) differ from those in the initial demonstration, thereby enabling a robust evaluation of the model's generalization ability.

### B.1.3 IMPLEMENTATION DETAILS OF SRLR MODULES

Each stage of the SRLR loop is implemented as a distinct prompt-based query to the high-level VLM ($\pi_H$). The specific prompts used for each module are shown below.

---

**Summarize module prompt**

```
Analyze the demonstration and extract the core operational strategy that
an AI agent can use to handle similar tasks.

Task: {task_goal}
Step: {instructions} -> {action_raws}

Output Requirements:
1.Logical Flow: Describe the steps in the exact original order using
First, Then, After that....
2.Critical Success Factors: Summarize 3 or 4 essential rules for success
3.UI Interaction Patterns: Highlight the main interaction methods

Notes:
Keep it concise, emphasize sequence and dependencies.
Generalize the technique rather than focusing on specific content.
```

---

**Reflect module prompt**

```
Compare and analyze the main error reason.

Error Trajectory: {error_trajectory}
Correct Demonstration: {demo_trajectory}

Analysis Requirements:
1.Compare the sequence step by step against the demonstration to identi-
fy where divergence occurs.
2.Locate the failure within the five information-handling levels:
(1) Source Location
(2) Target Selection
(3) Data Extraction
(4) Data Processing
(5) Answer Output
3.Distinguish between wrong target selection and failure in extraction
or processing within the correct target.
4.Provide evidence from the demonstration to show what information was
missing, incorrect, or not properly handled.

Output Format:
ERROR_REASON: State the exact failure point and specify the missing or
incorrect information.
EXPLANATION: Provide step-by-step comparison showing where the error tr-
ajectory diverged from the demonstration and support it with evidence.
```

---

**Locate module prompt**

```
Conduct a step-by-step process to pinpoint where the error occurred,
using error details and demonstration data as references.

Inputs:
Error Reason: {error_reason_72b}
Demonstration Steps: {demonstration}
Prompt with Line Numbers: {prompt_line}
```

---

```
Process:
Step 1 : Error Location
- Review the stated error reason: {error_reason_72b}
- Identify the exact line numbers in the prompt linked to this error
- Highlight the sections of content that contributed to the issue
Step 2 : Demonstration Reference
- Match the error against relevant demonstration steps
- Extract the instruction that address the error cause
- Specify which part of the demonstration resolves the issue

Output Format:
STEP1_ERROR_LOCATION: [Line number of the first failed step and related
content in the prompt connected to the error]
STEP2_DEMO_CONTENT_LOCATION: [Relevant demonstration steps (instruction
only) that address the error]
```

**Revise module prompt**

```
Modify the prompt based on error analysis and demonstration data.

Inputs
Demonstration Steps: {demonstration}
Prompt Used in Failed Execution: {prompt_line}
Error Location Analysis: {error_location_analysis}

Improvement Process
Step 1 : Direct Prompt Modification
- Emphasize key attention points and critical considerations rather than
adding examples.
- Highlight critical notices using markers such as "IMPORTANT:", "CRI-
TICAL:", "NOTE:", or "PAY ATTENTION TO:".
- Use demonstration reasoning to guide attention points without copying
specific results.
- Treat demonstration data as the source of truth; only include functio-
nality shown in the demonstration.
Step 2 : Semantic Alignment Check
- Review each line of the modified prompt for semantic consistency with
demonstration content.
- For each line, identify the corresponding demonstration instruction
and ensure it aligns in meaning.
- Remove any content not supported by demonstration data.

Focus on:
- Precision: Target exact locations and content.
- Demonstration-Based: All modifications must be grounded in demonstrat-
ion steps.
- Attention Emphasis: Highlight critical points.
- Semantic Consistency: Ensure meaning matches demonstration instruction
- Functionality Verification: Only include demonstrated features.

Output Format
FINAL_MODIFIED_PROMPT: [Complete modified prompt after both steps, with
every line supported by demonstration data]
```

### B.1.4   ANALYSIS OF INDUCED DECLARATIVE KNOWLEDGE AND REFLECTION ROBUSTNESS

To further understand the behavior and limitations of the SRLR loop, we conducted a comprehensive analysis of the induced knowledge bases ($K_G$) and the corresponding reflection logs. This subsection summarizes our observations, and the next section provides extended case studies and visualizations.

**(1) Taxonomy of Induced Declarative Knowledge.** Across AndroidWorld tasks, the Summarize stage consistently induces four primary categories of declarative knowledge (distribution shown in Figure 8):

- **Step Ordering**: Core logical dependencies between task steps, such as "Open the Audio Recorder app before accessing the file list."
- **UI Layout & Invariants**: Stable visual semantics of UI elements, e.g., "The record button is the white circle at the bottom center."
- **Parameter Constraints**: Format requirements and mandatory input rules, such as "The filename must include the '.m4a' extension."
- **Recovery Strategies**: Conditional checks for resolving common execution anomalies, e.g., "If the 'Save' button is disabled, ensure the text field is focused."

These types capture task-level logic that is naturally expressible in language and form the planner's declarative backbone.

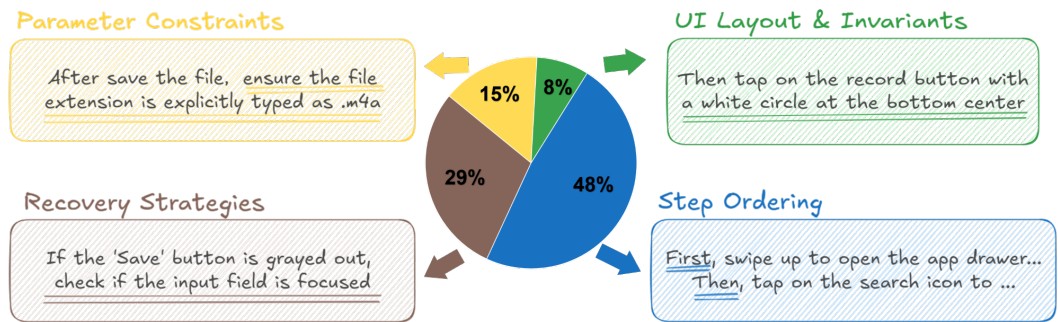

Figure 8: Distribution of the induced declarative knowledge

**(2) Boundaries of Summarizable Knowledge.** We also identified knowledge types that are inherently difficult to verbalize. These limitations highlight the need for the procedural low-level executor:

- **Ineffable Visual Grounding**: Precise spatial relations or pixel-level cues that lack stable linguistic descriptions (e.g., "Tap near the 3 o'clock position").
- **Visual Dynamics**: Behaviors that require continuous perceptual feedback, such as iterative swiping until a list terminus is reached.
- **Massive Episodic Content**: Tasks involving retrieval from large historical content, e.g., searching through dozens of previously viewed images.

While our framework focuses on evolving task logic rather than maintaining long-range episodic memory, each of these limitations points to directions for complementary improvements. First, *ineffable visual grounding* could be mitigated by stronger backbone vision encoders or multimodal pretraining that yields more precise spatial understanding. Second, *visual dynamics* may be addressed by integrating recurrent perceptual modules or short-horizon visual predictors capable of modeling iterative feedback. Third, *massive episodic content* would require external memory systems or retrieval-augmented modules, which are orthogonal to the SRLR loop but could complement it in future extensions.

**(3) Self-Correction of Incorrect Declarative Knowledge.** In some cases, the initial Summarize stage induces overly specific or incorrect rules, especially when a demonstration contains instance-specific details. The SRLR loop naturally corrects such issues: *Reflect* identifies the state mismatch, *Locate* isolates the faulty rule, and *Revise* updates and generalizes it.

For example, in the AudioRecorder task (the next subsection), the initial knowledge incorrectly memorized a fixed filename. After an execution failure, the rule was revised to "Type the filename specified in the current instruction," leading to consistent generalization across future episodes.

**(4) Scope and Limitations of Root-Cause Analysis.** The reflection process effectively captures failures that leave clear visual evidence, including:

- omitted or mis-ordered steps,

- incorrect UI navigation,

- wrong element selection or misclicks,

- invalid textual inputs or wrong parameter values.

However, reflection is fundamentally limited by visual observability. Certain execution failures do not manifest in screenshots, such as: OS-level freezes or dropped touch events, network delays preventing UI updates and invisible permission denials or background system interrupts.

These cases yield no visual divergence, and thus no reliable root-cause attribution. To maintain robustness under such conditions, the system employs a strict *max-retry* fallback mechanism (Algorithm 1, line [3]) to avoid infinite loops.

### B.1.5 EVOLUTION OF THE KNOWLEDGE BASE ($K_G$)

To concretely illustrate the self-evolution process of the declarative knowledge base ($K_G$) via the SRLR loop, we showcase its revision process for the **'AudioRecorderRecordAudioWithFile-Name'** task. The evolution from $K_G^0 \rightarrow K_G^1 \rightarrow K_G^2$ demonstrates a sophisticated learning pattern: it begins with a literal summary, evolves to a generalized and logically structured plan, and finally refines into a robust strategy that re-introduces critical, grounded details discovered through further interaction.

**Initial Knowledge ($K_G^0$) from `Summarize` Phase.** The process starts with the `Summarize` module, which distills a single expert demonstration into an initial, structured knowledge base, $K_G^0$. This plan is a direct, flat list of actions observed in the demonstration.

---

**$K_G^0$: Initial Plan from Demonstration**

- Swipe up on the screen to reveal more apps.
- Tap on the "Audio Recorder" app icon.
- Tap the white circular button at the bottom of the screen to start recording.
- Tap the white square button at the bottom right of the screen to stop the recording.
- Long press the backspace key on the keyboard to delete the content in the input field.
- Type the text "presentation_fGwr.m4a".
- Tap the "Save" button.

---

**SRLR Cycle 1: Generalization and Logical Structuring.** During execution, the agent fails on a new task instance due to the hard-coded filename. The SRLR loop is triggered.

- **`Reflect`**: The agent observes the failure and identifies other potential ambiguities, such as implicit preconditions.

- **`Locate`**: The failure's root cause is traced to the specific filename and the plan's lack of explicit logical flow and error-checking.

- **`Revise`**: The knowledge is updated in two major ways: (1) It's **generalized** by replacing the hard-coded name with a placeholder. (2) It's reformed into a more robust, **logically structured** sequence with explicit steps for verification and safeguards (e.g., "Ensure no other actions are taken...").

---

**$K_G^1$: Revision 1 (Generalization & Logical Structure)**

- ~~Swipe up... *and* Tap on...~~
- First, you need to locate and open the Audio Recorder app. *(Abstracted)*
- ~~Tap the white circular button...~~
- To start recording, tap the record button. Ensure no other actions are taken until the recording begins. *(Added safeguard)*
- ~~Long press the backspace key...~~ *(Removed as potentially over-specific)*
- ~~Type the text "presentation_fGwr.m4a".~~
- Then, navigate to the save options... and enter `[FILENAME]`... *(Generalized)*
- Following this, ensure the file type is set to `.m4a`... *(New step from failure analysis)*
- Before confirming..., verify that both the file name and type are correctly entered... *(Added verification step)*

---

**SRLR Cycle 2: Re-introducing Critical Specificity and Grounding.** While $K_G^1$ is more logical, its abstraction causes new failures. The agent struggles with grounding (e.g., finding the generic "record button") and, critically, fails to clear default text in the input field.

- **Reflect**: The agent recognizes that filenames are consistently corrupted (e.g., 'Recording1[FILENAME].m4a') and that it sometimes hesitates or clicks the wrong UI element.
- **Locate**: The root causes are identified: (1) the omission of the crucial step to clear the input field and (2) over-abstraction of UI element descriptions.
- **Revise**: The plan is refined to achieve a balance. It **re-introduces critical actions** as mandatory preconditions ("you **MUST** long press...") and **restores specific UI details** ("white circular button") for better grounding, while retaining the strong logical flow.

---

**$K_G^2$: Revision 2 (Robust and Re-grounded Plan)**

- ~~First, you need to locate and open the Audio Recorder app.~~
- First, swipe up on the screen... and Then, tap on the "Audio Recorder" app icon. *(Re-introduced specificity)*
- ~~To start recording, tap the record button. Ensure...~~
- After opening the app, tap the **white circular button**...
- When you're ready to stop..., tap the **white square button**... *(Restored UI details for grounding)*
- ~~*(Verification and file type steps from $K_G^1$)*~~
- Before typing the filename, to delete any existing content, you **MUST** long press the backspace key... *(Critical step re-introduced as mandatory)*
- Type `[FILENAME]` into the designated input field...
- Tap the "Save" button to finalize...

---

This final knowledge base, $K_G^2$, is superior to its predecessors: it is general enough to handle different task parameters (like $K_G^1$) yet specific enough to execute robustly and avoid common pitfalls (like $K_G^0$). This demonstrates the effectiveness of the SRLR loop in creating a robust, reusable knowledge base from minimal initial data.

### B.2 LOW-LEVEL EXECUTOR: C-GRPO IMPLEMENTATION

#### B.2.1 PSEUDOCODE FOR THE C-GRPO ALGORITHM

The C-GRPO algorithm trains the low-level executor ($\pi_L$) by combining a curriculum learning strategy with the Group Relative Policy Optimization framework. Algorithm 2 provides a high-level overview of the training process. It begins by initializing the error-decoupled replay buffers based

---

**Algorithm 2:** Curriculum-Guided Group Relative Policy Optimization (C-GRPO)

**Input** : Initial policy $\pi_L(\theta)$, Expert demonstration dataset $\mathcal{D}_{\text{expert}}$
**Output** : Trained policy parameters $\theta$

  // Phase 1: Initialize Error-Decoupled Replay Buffers
1 Initialize replay buffers $\mathcal{D}_{\text{con}}, \mathcal{D}_{\text{type}}, \mathcal{D}_{\text{param}}$
2 **foreach** *sample* $i = (s_i, \hat{a}_i, \mathcal{T}_i^d)$ *in* $\mathcal{D}_{\text{expert}}$ **do**
3      Calculate error rates $\eta_{\text{type}}(i), \eta_{\text{param}}(i)$ with the initial policy $\pi_L$
4      Assign sample $i$ (along with its error rates) to the corresponding buffer based on thresholds

  // Phase 2: Main Training Loop
5 **for** $k \leftarrow 1$ **to** $K_{\max}$ **do**
    // Construct a balanced mini-batch
6      Sample mini-batch $\mathcal{B}$ from buffers $\mathcal{D}_{\text{con}}, \mathcal{D}_{\text{type}}, \mathcal{D}_{\text{param}}$ according to ratios $\{\beta\}$

    // Augment each sample with Dynamic Demonstration Injection
7      **foreach** *sample* $i = (s_i, \hat{a}_i, \mathcal{T}_i^d, \eta_{...})$ *in* $\mathcal{B}$ **do**
8          $d_i \leftarrow \eta_{\text{type}}(i) + \eta_{\text{param}}(i)$         // Calculate sample difficulty
9          $l_i \leftarrow |\mathcal{T}_i^d| \cdot \sigma(k) \cdot f_{\text{gate}}(d_i)$       // Calculate injected prefix length
10         $c_i \leftarrow \text{GetPrefix}(\mathcal{T}_i^d, l_i) \oplus s_i$      // Form augmented context

    // Perform C-GRPO update on the augmented batch
11      Compute loss $\mathcal{J}_{\text{C-GRPO}}(\theta; \{c_i\})$   // Generate rollouts, compute rewards/advantages
12      $\theta \leftarrow \theta - \alpha\nabla_\theta\mathcal{J}_{\text{C-GRPO}}(\theta)$
13 **return** $\theta$

---

on the initial policy's performance. It then enters a main training loop where balanced mini-batches are constructed, augmented with dynamic demonstration prefixes, and used for the policy update, effectively guiding the agent towards acquiring robust procedural skills.

### B.2.2 TRAINING DATA CONSTRUCTION

The training dataset for the low-level executor, $\pi_L$, was constructed from the 116 high-quality, multi-step expert demonstration trajectories sourced from the AndroidWorld benchmark. To adapt this data for training a single-step action policy, we performed a meticulous preprocessing pipeline.

**Processing Pipeline.** First, each trajectory was decomposed into a sequence of single-step state-action pairs. Second, every pair underwent a manual verification process to ensure its quality and correctness. We filtered out any steps that were ambiguous or erroneous, such as actions that resulted in no observable change on the screen or where the natural language instruction did not precisely match the recorded atomic action. This rigorous cleaning process yielded a final dataset of **606 high-quality samples**.

**Data Distribution.** The dataset encompasses a wide variety of UI interactions across the 20 applications in AndroidWorld. Critically, the distribution of action types is naturally imbalanced, with common actions like `click` appearing far more frequently than less common but equally important actions such as `long_press` and `swipe`. This imbalance underscores the need for the error-decoupled replay balancing mechanism in our C-GRPO algorithm.

**Data Format.** Each sample was serialized into a JSON object, paired with its corresponding screenshot. The JSON structure is designed to be compatible with standard VLM training frameworks. An example is shown in Figure 9. The keys are defined as follows:

- `id`: A unique identifier for the data sample.
- `task`: The high-level goal of the entire trajectory.
- `conversations`: A list containing the human-like instruction (the sub-goal for the current step) and the ground-truth tool call from the GPT-like model (the expert action).

- `image`: The file path to the screenshot taken just before the action.

Figure 9: An example of a single training data sample in JSON format.

```
Training Sample Example

{
  "id": "action_MarkorMoveNote_step_0_20250825_164306",
  "task": "In Markor, move the note 8zum_friendly_penguin.txt
          from WorkProjects to CodeSnippets.",
  "conversations": [
    {
      "from": "human",
      "value": "<image>\nSwipe up on the screen to locate
              the Markor app in the app drawer."
    },
    {
      "from": "gpt",
      "value": "<tool_call>"arguments\": {"action": "swipe",
              "coordinate": [546, 2000],
              "coordinate2": [546, 800]}}\n
              </tool_call>"
    }
  ],
  "image": "images/screenshot_20250825_MarkorMoveNote_0000.png"
}
```

### B.2.3 HYPERPARAMETER SETTINGS

Our experimental setup involves distinct configurations for the high-level planner and the low-level executor. The planner operates in a training-free manner, guided by the hyperparameters of the SRLR loop. The executor is trained via our C-GRPO algorithm, which we implemented by adapting the `GRPOTrainer` framework from VLM-R1[1] (Shen et al., 2025). All key hyperparameters for each component, along with the training infrastructure details, are consolidated in Table 5.

---

[1]`https://github.com/om-ai-lab/VLM-R1`

Table 5: Comprehensive list of hyperparameters for K²-Agent.

| Component | Hyperparameter | Value |
|---|---|---|
| **High-Level Planner ($\pi_H$)** | | |
| MAX_ITER | Max revision cycles per task | 10 |
| SUCCESS_THRESH | Consecutive successes to stop | 3 |
| **Low-Level Executor ($\pi_L$)** | | |
| Base Learning Rate | Learning rate for policy | $1 \times 10^{-6}$ |
| Epoch Count | Number of training epochs | 2 |
| Per-GPU Batch Size | Samples per GPU during training | 2 |
| Generations per Input ($G$) | Number of rollouts per sample | 8 |
| Gradient Accumulation Steps | Steps to accumulate gradients | 2 |
| KL Weight ($\beta$) | Weight for KL penalty | 0.04 |
| Clipping Ratio ($\epsilon$) | GRPO clipping ratio | 0.2 |
| Reward Weights ($\lambda_{\text{fmt}}, \lambda_{\text{content}}$) | Weights for format/content reward | {1.0, 1.0} |
| **C-GRPO Curriculum Strategy** | | |
| Buffer Ratios ($\beta_{\text{con}}, \beta_{\text{type}}, \beta_{\text{param}}$) | Sampling ratios for replay pools | {0.5, 0.25, 0.25} |
| Injection Temp ($T$) | Temperature for difficulty gating | 0.5 |
| Max Training Steps ($K_{\text{max}}$) | Total steps for annealing scheduler | 1000 |
| **Model and Training Infrastructure** | | |
| High-Level Model ($\pi_H$) | VLM for planner | Qwen2.5-VL-72B |
| Low-Level Model ($\pi_L$) | VLM for executor | Qwen2.5-VL-7B |
| Max Input Tokens | Context length limit | 1024 |
| Max Output Tokens | Generation length limit | 256 |
| Optimizer Precision | Mixed-precision training type | `bfloat16` |
| Hardware | GPUs used for training | 8 x NVIDIA A100 (80GB) |

All experiments are carried out on a cluster of eight NVIDIA A100 GPUs with 80 GB memory each, and a complete training pass requires about eight hours. The software environment includes `flash_attn` 2.8.3, `torch` 2.8.0, `transformers` 4.49.0, and `trl` 0.17.0, which together enable efficient use of FlashAttention and smooth integration with the GRPOTrainer workflow.

# C  EXTENDED DISCUSSION AND COMPARATIVE ANALYSIS

This section elaborates on the distinctions between K²-Agent and other prominent exploration or hybrid frameworks, specifically discussing the applicability of Hindsight Experience Replay (HER) Andrychowicz et al. (2017) and contrasting our architecture with ReAct Yao et al. (2022), Voyager Wang et al. (2023), and RPA systems.

## C.1  INAPPLICABILITY OF HINDSIGHT EXPERIENCE REPLAY (HER)

While HER (Andrychowicz et al., 2017) is a powerful exploration technique for goal-conditioned RL, it relies on the assumption that any visited state can be re-labeled as a valid alternative goal. This assumption fundamentally conflicts with our vision-based, instruction-following setting. First, our "goals" are natural language instructions (e.g., *"Open Settings"*), while states are pixel-level screenshots; there is no trivial mapping to convert an arbitrary intermediate screen back into a high-level semantic instruction. Second, unlike robotic manipulation where reaching any coordinate is physically valid, many intermediate GUI states (e.g., loading screens or partial lists) do not correspond to meaningful user tasks, making goal re-labeling semantically undefined. Consequently, instead of goal re-labeling, we employ **Dynamic Demonstration Injection** as a domain-adapted curriculum to bootstrap exploration in this sparse-reward environment.

## C.2  COMPARISON WITH ALTERNATIVE HYBRID ARCHITECTURES

K²-Agent represents a distinct evolution from existing "reasoning + acting" frameworks. **(1) vs. ReAct:** Our "No Hierarchy" baseline (Table 4) mirrors a ReAct-style setup where a monolithic policy handles both reasoning and acting. The significant performance gap (35.3% vs. 76.1%)

confirms that explicitly decoupling "know-what" from "know-how" is critical for preventing cognitive drift in long-horizon GUI tasks. **(2) vs. Voyager:** While systems like Voyager evolve skills as executable code over structured APIs, such stable interfaces are unavailable in vision-only mobile control. $K^2$-Agent instead evolves skills as parametric neural policies via C-GRPO, enabling operation in pixel-based environments where code generation is inapplicable. **(3) vs. RPA:** Unlike Classical Robotic Process Automation (RPA) which relies on brittle, static scripts, $K^2$-Agent is data-driven. Its SRLR loop adaptively refines knowledge, and its executor generalizes zero-shot to unseen apps and platforms (as evidenced by AitW and ScreenSpot-v2 results), offering a robust alternative to fixed automation pipelines.

# D  EXPERIMENTAL SETUP AND ADDITIONAL RESULTS

## D.1  BENCHMARK AND EVALUATION DETAILS

### D.1.1  ENVIRONMENTS DETAILS

**Environment.** Our primary evaluation platform is AndroidWorld (Rawles et al., 2024), a standardized benchmark that operates on a live Android emulator. It features 116 hand-crafted tasks distributed across 20 diverse applications. Crucially, to test for generalization and prevent solution memorization, each task is dynamically instantiated with randomized parameters for every episode. The tasks span a wide range of complexities, from easy operations to long-horizon procedures. Table 6 provides a detailed breakdown of the applications and their corresponding task counts.

**Observation Space.** Our agent interacts with an Android Virtual Device (AVD) configured to emulate a Pixel 6 running Android Tiramisu (API Level 33), consistent with the standard setup for the AndroidWorld benchmark. We adopt a vision-centric approach where the agent's perception relies solely on raw visual input. The state representation at each step $t$ is a multimodal input comprising:

- **Screenshot**: An RGB image of the current screen with a resolution of $2400 \times 1080$ pixels. We do not use any underlying structural information, such as the accessibility tree (A11y tree) or view hierarchy XML, making the task more reliant on the model's visual understanding.
- **Task Goal**: A natural language string describing the overall objective, for example, "Record an audio clip using Audio Recorder app and save it."
- **History**: The sequence of past actions taken within the episode. This historical context is provided exclusively to the high-level planner ($\pi_H$) to support multi-step reasoning and error analysis in the SRLR loop. The low-level executor ($\pi_L$) operates without this history, focusing only on executing the current subgoal based on the present visual state.

**Action Space.** To facilitate robust and precise interaction with the mobile device environment, we define a structured action space for the low-level executor. Inspired by function-calling APIs, this design decouples the agent's intent into discrete action types and their corresponding parameters. This approach simplifies the learning task for the procedural model, allowing it to focus on grounding high-level subgoals to specific, executable operations. Table 7 provides a comprehensive summary of each action, its parameters, and its operational description within our framework.

**Reward Design.** Our reward design distinguishes between the training-free high-level planner and the learning-based low-level executor. The high-level planner, $\pi_H$, does not optimize its parameters via reward signals; instead, it uses the sparse, binary task-completion feedback from the environment solely to trigger its SRLR self-evolution loop.

The training of the low-level executor, $\pi_L$, is guided by a dense, step-wise composite reward signal $R_t$. This signal is designed to provide fine-grained feedback on the quality of the generated actions and is composed of two key components: a format reward and a content reward.

**Format Reward ($r_{\text{fmt}}$).** This reward component ensures that the model's output strictly adheres to our predefined tool-calling schema. An action can only be parsed and executed by the environment if it is formatted correctly. We define this as a binary indicator:

$$r_{\text{fmt}}(o_t) = \mathbb{1}\left\{o_t \text{ correctly matches the } \texttt{<tool\_call>}\{...\}\texttt{</tool\_call>} \text{ schema}\right\},$$

where $o_t$ is the raw text output generated by the model at step $t$. A reward of 1 is given for a valid format, and 0 otherwise.

Table 6: Overview of the 20 applications and the number of associated tasks in the Android-World(Rawles et al., 2024). The description for each app highlights its core functionality.

| | Application | Description | # Tasks |
|---|---|---|---|
| | Simple Calendar Pro | A calendar app for creating, deleting, and managing events and appointments. | 17 |
| | Settings | The Android system settings app for managing device settings such as Bluetooth, Wi-Fi, and brightness. | 15 |
| | Markor | A note-taking app for creating, editing, deleting, and managing notes and folders. | 14 |
| | Broccoli - Recipe App | A recipe management app for adding, deleting, and organizing recipes. | 13 |
| | Pro Expense | An expense tracking app for adding, deleting, and managing expenses. | 9 |
| | Simple SMS Messenger | An SMS app for sending, replying to, and re-sending text messages. | 7 |
| | OpenTracks | A sport tracking app for recording and analyzing activities, durations, and distances. | 6 |
| | Tasks | A task management app for tracking tasks, due dates, and priorities. | 6 |
| | Clock | An app with stopwatch and timer functionality. | 4 |
| | Joplin | A note-taking app. | 4 |
| | Retro Music | A music player app. | 4 |
| | Simple Gallery Pro | An app for viewing images. | 4 |
| | Camera | An app for taking photos and videos. | 3 |
| | Chrome | A web browser app. | 3 |
| | Contacts | An app for managing contact information. | 3 |
| | OsmAnd | A maps and navigation app with support for adding location markers, favorites, and saving tracks. | 3 |
| | VLC | A media player app for playing media files. | 3 |
| | Audio Recorder | An app for recording and saving audio clips. | 2 |
| | Files | A file manager app for the Android filesystem, used for deleting and moving files. | 2 |
| | Simple Draw Pro | A drawing app for creating and saving drawings. | 1 |

**Content Reward ($r_{\text{content}}$).** Given a correctly formatted output, this component evaluates the operational correctness of the action $a_t$ by comparing it to the ground-truth expert action $\hat{a}_t$. The reward assesses both the chosen action type (e.g., `click` vs. `swipe`) and the precision of its parameters (e.g., coordinates). The content reward is defined as:

$$r_{\text{content}}(a_t, \hat{a}_t) = \mathbb{1} \left\{ \begin{array}{l} \text{type}(a_t) = \text{type}(\hat{a}_t) \ \wedge \\ \|\text{param}(a_t) - \text{param}(\hat{a}_t)\| < \epsilon \end{array} \right\},$$

Table 7: The structured action space of K²-Agent. Each action is defined as a function call with specific arguments to control the mobile device.

| Action | Description | Arguments |
|---|---|---|
| click | Performs a standard, short tap on a specific screen location. Used for activating buttons, selecting items, or placing cursor focus. | *coordinate* |
| long_press | Executes a sustained press at a given coordinate. Essential for tasks like selecting text, opening context menus, or revealing hidden options. | *coordinate* |
| swipe | Drags from a starting point to an ending point. Used for scrolling, navigating pages, or adjusting sliders. | *coordinate, coordinate2* |
| type | Inputs a character sequence into the currently focused text field. This action directly injects text, bypassing the on-screen keyboard. | *text* |
| system_button | Triggers a system-level hardware button command, such as navigating back or returning to the home screen. | *button* |
| terminate | Ends the current task episode, reporting the final outcome. This signals to the high-level planner whether the goal was achieved. | *status* |
| answer | Provides a natural language response. This action is specifically used for information retrieval tasks where the goal is to find and report information rather than manipulate the UI. | *response_text* |

where $\text{type}(\cdot)$ returns the action's type, and $\text{param}(\cdot)$ extracts its parameters. For coordinate-based actions, the norm $\|\cdot\|$ is the Euclidean distance ($L_2$), while for text-based actions, it corresponds to an exact string match. The tolerance threshold $\epsilon$ is used for coordinate matching.

**Total Reward.** The final reward for training $\pi_L$ is a weighted combination of the two components:

$$R_t = \lambda_{\text{fmt}} \cdot r_{\text{fmt}}(o_t) + \lambda_{\text{content}} \cdot r_{\text{content}}(a_t, \hat{a}_t).$$

In our implementation, we set $\lambda_{\text{fmt}} = 1.0$ and $\lambda_{\text{content}} = 1.0$. Empirically, this balanced weighting enables the 7B executor to achieve a favorable trade-off among format compliance, grounding accuracy, and convergence speed.

## D.2 EXTENDED QUANTITATIVE ANALYSIS

### D.2.1 ANDROIDWORLD LEADERBOARD SNAPSHOT

We provide here an anonymous snapshot of the official *AndroidWorld* leaderboard as of August 2025. Figure 10 shows the ranking of our **K²-Agent**, which achieves a success rate of **76.7%**, placing **1st among all methods that rely solely on raw screenshots and open-source backbones**. For fairness and compliance with the double-blind review process, our submission was made through an anonymous GitHub repository and contains no identifying information. Competing systems that leverage additional privileged inputs (e.g., the accessibility tree) or closed-source backbones are also listed for reference.

### D.2.2 DETAILED PERFORMANCE STATISTICS

To provide a granular view of our agent's performance and facilitate detailed comparisons, Table 8 presents the success/failure outcome for K²-Agent and several key baselines on every one of the

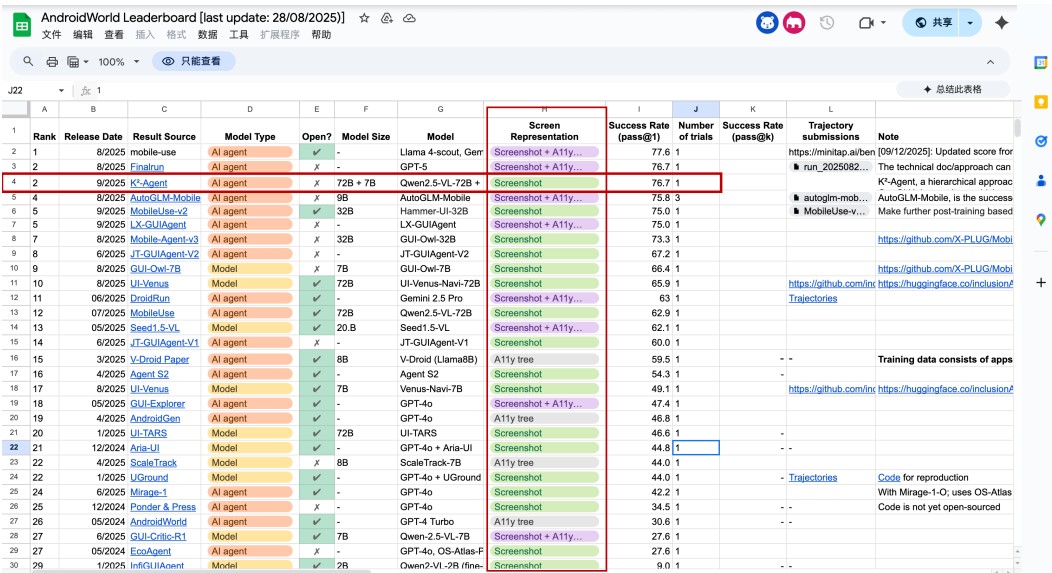

Figure 10: Public leaderboard of AndroidWorld as of August 2025, showing K²-Agent ranked 1st among all methods using only raw screenshots.

116 tasks in the AndroidWorld benchmark. This table also includes the results from our declarative knowledge transfer experiments, showing the performance of K²-Agent when its high-level planner is replaced with Gemini and GPT-4o backbones using the same evolved knowledge base. This allows for a direct, task-by-task assessment of where different models excel or fall short.

Table 8: Detailed performance comparison on all 116 tasks in AndroidWorld.

| Task | K²-Agent | mobile-use | K²-Agent(Gemini) | UI-Venus-72B | DroidRun | K²-Agent(Gpt-4o) |
|---|---|---|---|---|---|---|
| AudioRecorderRecordAudio | ✓ | ✓ | ✓ | ✓ | ✓ | ✗ |
| AudioRecorder-FileName | ✓ | ✓ | ✓ | ✗ | ✓ | ✓ |
| BrowserDraw | ✗ | ✗ | ✗ | ✗ | ✗ | ✗ |
| BrowserMaze | ✓ | ✓ | ✗ | ✗ | ✓ | ✗ |
| BrowserMultiply | ✓ | ✓ | ✓ | ✗ | ✓ | ✓ |
| CameraTakePhoto | ✓ | ✓ | ✓ | ✓ | ✓ | ✓ |
| CameraTakeVideo | ✓ | ✓ | ✓ | ✓ | ✓ | ✓ |
| ClockStopWatchPausedVerify | ✓ | ✓ | ✓ | ✓ | ✓ | ✓ |
| ClockStopWatchRunning | ✓ | ✓ | ✓ | ✓ | ✓ | ✓ |
| ClockTimerEntry | ✓ | ✓ | ✓ | ✓ | ✓ | ✗ |
| ContactsAddContact | ✓ | ✓ | ✓ | ✓ | ✓ | ✗ |
| ContactsNewContactDraft | ✓ | ✓ | ✗ | ✓ | ✓ | ✗ |
| ExpenseAddMultiple | ✓ | ✓ | ✓ | ✓ | ✓ | ✓ |
| ExpenseAddMultipleFromGallery | ✗ | ✓ | ✗ | ✗ | ✗ | ✗ |
| ExpenseAddMultipleFromMarkor | ✗ | ✗ | ✗ | ✗ | ✗ | ✗ |
| ExpenseAddSingle | ✓ | ✓ | ✓ | ✓ | ✓ | ✓ |
| ExpenseDeleteDuplicates | ✓ | ✓ | ✗ | ✓ | ✓ | ✓ |
| ExpenseDeleteDuplicates2 | ✓ | ✓ | ✓ | ✓ | ✓ | ✗ |
| ExpenseDeleteMultiple | ✓ | ✓ | ✓ | ✓ | ✓ | ✓ |
| ExpenseDeleteMultiple2 | ✓ | ✓ | ✗ | ✓ | ✓ | ✓ |

Continued on next page

Table 8 – continued from previous page

| Task | K²-Agent | mobile-use | K²-Agent(Gemini) | UI-Venus-72B | DroidRun | K²-Agent(Gpt-4o) |
|---|---|---|---|---|---|---|
| ExpenseDeleteSingle | ✓ | ✓ | ✓ | ✓ | ✓ | ✓ |
| FilesDeleteFile | ✓ | ✓ | ✓ | ✗ | ✓ | ✓ |
| FilesMoveFile | ✓ | ✓ | ✓ | ✗ | ✓ | ✓ |
| MarkorAddNoteHeader | ✗ | ✗ | ✗ | ✗ | ✗ | ✗ |
| MarkorChangeNoteContent | ✓ | ✗ | ✓ | ✗ | ✓ | ✓ |
| MarkorCreateFolder | ✓ | ✓ | ✓ | ✓ | ✓ | ✓ |
| MarkorCreateNote | ✓ | ✓ | ✓ | ✓ | ✓ | ✓ |
| MarkorCreateNoteAndSms | ✓ | ✗ | ✓ | ✗ | ✗ | ✗ |
| MarkorCreateNoteFromClipboard | ✓ | ✓ | ✓ | ✗ | ✗ | ✗ |
| MarkorDeleteAllNotes | ✓ | ✓ | ✓ | ✓ | ✗ | ✓ |
| MarkorDeleteNewestNote | ✓ | ✓ | ✗ | ✓ | ✓ | ✗ |
| MarkorDeleteNote | ✓ | ✓ | ✓ | ✓ | ✓ | ✓ |
| MarkorEditNote | ✓ | ✓ | ✓ | ✓ | ✗ | ✗ |
| MarkorMergeNotes | ✗ | ✗ | ✗ | ✗ | ✗ | ✗ |
| MarkorMoveNote | ✓ | ✓ | ✗ | ✗ | ✗ | ✓ |
| MarkorTranscribeReceipt | ✗ | ✗ | ✓ | ✗ | ✗ | ✗ |
| MarkorTranscribeVideo | ✗ | ✗ | ✗ | ✗ | ✗ | ✗ |
| NotesIsTodo | ✓ | ✓ | ✓ | ✓ | ✓ | ✗ |
| NotesMeetingAttendeeCount | ✓ | ✓ | ✓ | ✓ | ✓ | ✓ |
| NotesRecipeIngredientCount | ✓ | ✓ | ✓ | ✓ | ✗ | ✓ |
| NotesTodoItemCount | ✓ | ✓ | ✓ | ✓ | ✓ | ✓ |
| OpenAppTaskEval | ✓ | ✓ | ✓ | ✓ | ✓ | ✓ |
| OsmAndFavorite | ✓ | ✓ | ✓ | ✓ | ✓ | ✓ |
| OsmAndMarker | ✓ | ✓ | ✓ | ✗ | ✗ | ✓ |
| OsmAndTrack | ✗ | ✗ | ✗ | ✗ | ✗ | ✗ |
| RecipeAddMultipleRecipes | ✗ | ✓ | ✓ | ✓ | ✓ | ✗ |
| RecipeAdd-FromImage | ✗ | ✓ | ✗ | ✗ | ✗ | ✗ |
| RecipeAdd-FromMarkor | ✗ | ✓ | ✗ | ✗ | ✗ | ✗ |
| RecipeAdd-FromMarkor2 | ✗ | ✗ | ✗ | ✗ | ✗ | ✗ |
| RecipeAddSingleRecipe | ✓ | ✓ | ✓ | ✓ | ✓ | ✓ |
| RecipeDeleteDuplicateRecipes | ✓ | ✓ | ✗ | ✓ | ✓ | ✓ |
| RecipeDeleteDuplicateRecipes2 | ✗ | ✗ | ✗ | ✗ | ✗ | ✗ |
| RecipeDeleteDuplicateRecipes3 | ✗ | ✗ | ✗ | ✗ | ✗ | ✗ |
| RecipeDeleteMultipleRecipes | ✓ | ✓ | ✓ | ✓ | ✓ | ✓ |
| RecipeDeleteMultiple-Constraint | ✗ | ✓ | ✗ | ✓ | ✗ | ✗ |
| RecipeDelete-WithNoise | ✓ | ✓ | ✓ | ✓ | ✓ | ✓ |
| RecipeDeleteSingleRecipe | ✓ | ✓ | ✓ | ✓ | ✓ | ✓ |
| RecipeDelete-WithNoise | ✓ | ✓ | ✓ | ✓ | ✓ | ✓ |
| RetroCreatePlaylist | ✓ | ✗ | ✓ | ✓ | ✓ | ✓ |
| RetroPlayingQueue | ✗ | ✗ | ✓ | ✓ | ✓ | ✗ |
| RetroPlaylistDuration | ✗ | ✗ | ✗ | ✗ | ✗ | ✗ |
| RetroSavePlaylist | ✗ | ✗ | ✓ | ✗ | ✗ | ✗ |
| SaveCopyOfReceiptTaskEval | ✓ | ✓ | ✓ | ✓ | ✓ | ✓ |
| SimpleCalendarAddOneEvent | ✗ | ✓ | ✗ | ✓ | ✓ | ✓ |
| SimpleCalendar-InTwoWeeks | ✗ | ✓ | ✓ | ✗ | ✗ | ✗ |
| SimpleCalendar-RelativeDay | ✓ | ✓ | ✓ | ✓ | ✗ | ✗ |
| SimpleCalendar-Tomorrow | ✓ | ✓ | ✗ | ✗ | ✓ | ✗ |
| SimpleCalendarAddRepeatingEvent | ✓ | ✓ | ✓ | ✓ | ✗ | ✗ |
| SimpleCalendarAnyEventsOnDate | ✓ | ✓ | ✓ | ✓ | ✗ | ✗ |
| SimpleCalendarDeleteEvents | ✓ | ✓ | ✓ | ✓ | ✓ | ✓ |
| SimpleCalendar-OnRelativeDay | ✓ | ✓ | ✓ | ✓ | ✓ | ✗ |
| SimpleCalendarDeleteOneEvent | ✓ | ✓ | ✓ | ✓ | ✗ | ✓ |
| SimpleCalendarEventOnDateAtTime | ✓ | ✓ | ✓ | ✓ | ✓ | ✓ |
| SimpleCalendarEventsInNextWeek | ✓ | ✗ | ✓ | ✗ | ✗ | ✓ |
| SimpleCalendarEventsInTimeRange | ✓ | ✓ | ✓ | ✓ | ✓ | ✗ |

Table 8 – continued from previous page

| Task | K²-Agent | mobile-use | K²-Agent(Gemini) | UI-Venus-72B | DroidRun | K²-Agent(Gpt-4o) |
|---|---|---|---|---|---|---|
| SimpleCalendarEventsOnDate | ✓ | ✓ | ✓ | ✓ | ✓ | ✓ |
| SimpleCalendarFirst-StartTime | ✓ | ✓ | ✓ | ✓ | ✓ | ✓ |
| SimpleCalendarLocationOfEvent | ✗ | ✓ | ✓ | ✓ | ✓ | ✗ |
| SimpleCalendarNextEvent | ✓ | ✓ | ✓ | ✗ | ✓ | ✓ |
| SimpleCalendar-WithPerson | ✓ | ✓ | ✓ | ✓ | ✗ | ✗ |
| SimpleDrawProCreateDrawing | ✓ | ✓ | ✓ | ✗ | ✓ | ✓ |
| SimpleSmsReply | ✓ | ✓ | ✓ | ✓ | ✗ | ✓ |
| SimpleSmsReplyMostRecent | ✓ | ✓ | ✓ | ✓ | ✓ | ✓ |
| SimpleSmsResend | ✓ | ✓ | ✓ | ✓ | ✓ | ✓ |
| SimpleSmsSend | ✓ | ✓ | ✓ | ✓ | ✗ | ✗ |
| SimpleSmsSendClipboardContent | ✓ | ✓ | ✓ | ✓ | ✗ | ✓ |
| SimpleSmsSendReceivedAddress | ✓ | ✓ | ✓ | ✓ | ✗ | ✓ |
| SportsTracker-ForWeek | ✓ | ✗ | ✓ | ✗ | ✗ | ✗ |
| SportsTrackerActivitiesOnDate | ✗ | ✗ | ✓ | ✗ | ✗ | ✓ |
| SportsTrackerActivityDuration | ✓ | ✓ | ✓ | ✓ | ✓ | ✓ |
| SportsTracker-Activity | ✓ | ✗ | ✓ | ✗ | ✓ | ✗ |
| SportsTrackerTotalDistance | ✓ | ✗ | ✗ | ✗ | ✓ | ✓ |
| SportsTrackerTotalDuration | ✗ | ✗ | ✓ | ✓ | ✗ | ✗ |
| SystemBluetoothTurnOff | ✓ | ✓ | ✓ | ✓ | ✓ | ✓ |
| SystemBluetoothTurnOffVerify | ✓ | ✓ | ✗ | ✓ | ✓ | ✓ |
| SystemBluetoothTurnOn | ✓ | ✓ | ✓ | ✓ | ✓ | ✓ |
| SystemBluetoothTurnOnVerify | ✓ | ✓ | ✓ | ✓ | ✓ | ✓ |
| SystemBrightnessMax | ✓ | ✓ | ✓ | ✓ | ✓ | ✓ |
| SystemBrightnessMaxVerify | ✓ | ✓ | ✓ | ✓ | ✓ | ✓ |
| SystemBrightnessMin | ✓ | ✓ | ✓ | ✓ | ✓ | ✓ |
| SystemBrightnessMinVerify | ✓ | ✓ | ✓ | ✓ | ✓ | ✓ |
| SystemCopyToClipboard | ✓ | ✓ | ✓ | ✗ | ✗ | ✓ |
| SystemWifiTurnOff | ✓ | ✓ | ✓ | ✓ | ✓ | ✓ |
| SystemWifiTurnOffVerify | ✓ | ✓ | ✗ | ✓ | ✓ | ✓ |
| SystemWifiTurnOn | ✓ | ✓ | ✓ | ✓ | ✓ | ✓ |
| SystemWifiTurnOnVerify | ✓ | ✓ | ✓ | ✓ | ✓ | ✓ |
| TasksCompletedTasksForDate | ✗ | ✗ | ✓ | ✗ | ✗ | ✓ |
| TasksDueNextWeek | ✗ | ✗ | ✗ | ✗ | ✗ | ✗ |
| TasksDueOnDate | ✓ | ✗ | ✗ | ✓ | ✗ | ✗ |
| TasksHighPriorityTasks | ✓ | ✗ | ✗ | ✓ | ✗ | ✗ |
| TasksHighPriorityTasksDueOnDate | ✓ | ✗ | ✓ | ✗ | ✗ | ✗ |
| TasksIncompleteTasksOnDate | ✓ | ✗ | ✓ | ✓ | ✓ | ✓ |
| TurnOffWifiAndTurnOnBluetooth | ✓ | ✓ | ✓ | ✓ | ✓ | ✓ |
| TurnOnWifiAndOpenApp | ✓ | ✓ | ✗ | ✓ | ✓ | ✓ |
| VlcCreatePlaylist | ✗ | ✗ | ✗ | ✗ | ✓ | ✗ |
| VlcCreateTwoPlaylists | ✗ | ✗ | ✗ | ✗ | ✓ | ✗ |
| **Success Rate (%)** | **76.7** | **74.1** | **72.4** | **65.9** | **62.9** | **58.6** |

# E  QUALITATIVE ANALYSIS AND CASE STUDIES

## E.1  GENERALIZATION CASE STUDIES

### E.1.1  DECLARATIVE KNOWLEDGE TRANSFER ACROSS BACKBONES

To validate the model-agnostic nature of the declarative knowledge ($K_G$), we transferred the final knowledge bases evolved by the Qwen2.5-VL-72B planner to new planners using Gemini-2.5-Pro and GPT-4o backbones. The transfer was conducted in a zero-shot setting, with the low-level executor held constant. The detailed per-task results are presented in Table 8. The aggregate success

rates—**72.4%** for the Gemini-based planner and **58.6%** for the GPT-4o-based planner—confirm that the declarative knowledge is fundamentally generalizable.

Our analysis reveals three further findings. First, the knowledge demonstrates essential portability; as shown in Figure 5(a), the core strategies distilled by SRLR are explicit and robust enough to improve performance across different model families. Second, an adaptation cost is evident in the performance gap between the original and new backbones. This is expected, as the linguistic phrasing and strategic details of the knowledge base were co-adapted through interaction with the Qwen model, and new models may incur a performance penalty when interpreting this tailored knowledge. Most notably, we observed the emergence of new capabilities. On several tasks where the original Qwen planner failed, such as `SimpleCalendarLocationOfEvent` and `SportsTrackerTotalDuration`, the planners using Gemini or GPT-4o succeeded with the *exact same* knowledge base. This highlights a powerful synergy where the transferred knowledge provides the correct high-level plan, and the superior intrinsic capabilities of the new backbone model enable it to successfully execute previously intractable steps.

### E.1.2 PROCEDURAL SKILL TRANSFER ACROSS BENCHMARKS

To comprehensively evaluate the generalization of learned procedural skills, we test the low-level executor, trained on AndroidWorld, on two distinct benchmarks in a zero-shot setting. We first use ScreenSpot-v2 to assess its fundamental, single-step grounding capabilities across platforms, and then use Android-in-the-Wild (AitW) to evaluate its applicability in complex, long-horizon tasks.

**Fundamental Grounding on ScreenSpot-v2.** To rigorously assess core GUI grounding capabilities, we conducted a transfer experiment on the complete **ScreenSpot-v2** benchmark (Wu et al., 2024). ScreenSpot-v2 is a reliable standard, containing tasks across three distinct domains: **Mobile** (iOS/Android), **Desktop** (Windows/macOS), and **Web**. The crucial aspect of this test is the domain mismatch: our executor, trained **exclusively on AndroidWorld (mobile) data**, was evaluated without any fine-tuning. This tests the hypothesis that the learned skills are fundamental enough to transfer not only to unseen mobile apps but also to entirely different operating paradigms.

As detailed in Table 9, K²-Agent's executor achieves a remarkable **91.3%** overall accuracy. This strong performance, despite the challenging cross-platform setting, validates our core hypothesis. By decoupling and focusing on procedural knowledge, the executor learns a robust, platform-agnostic visual grounding model rather than memorizing platform-specific patterns. This powerful generalization demonstrates that the executor's "muscle memory" is fundamentally about understanding visual language, making it a highly transferable component.

**Long-Horizon Application on Android-in-the-Wild.** Beyond single-step grounding, we also evaluated whether the learned procedural skills can be effectively chained to solve complex, multi-step tasks. We deployed the same executor, again without any fine-tuning, on the long-horizon tasks of the **Android-in-the-Wild** benchmark. For this experiment, the high-level planner was bootstrapped with a single expert demonstration for each task subset to rapidly generate a high-level plan.

As shown in Table 2, K²-Agent achieves state-of-the-art performance, with success rates of **86.5%** on AitW-General and **68.3%** on AitW-WebShopping. This significantly surpasses prior methods based on SFT, RL, or closed-source models, confirming that the robust procedural skills learned on AndroidWorld serve as a strong foundation for solving complex, unseen tasks.

### E.2 TRAJECTORY VISUALIZATIONS

To provide a more intuitive understanding of our agent's behavior, this section presents visual case studies of both successful and failed trajectories.

### E.2.1 SUCCESSFUL TRAJECTORY ON A COMPLEX TASK

Figure 11 illustrates complete steps from a successful execution of the complex, multi-app task `MarkorCreateNoteAndSms`. This trajectory demonstrates the effective synergy between the high-level planner and the low-level executor. At each stage, the planner, guided by its evolved

Table 9: Zero-shot performance of the K²-Agent executor on the ScreenSpot-v2 benchmark, compared to other state-of-the-art GUI grounding models. Our executor demonstrates competitive performance, highlighting the strong generalization of its learned procedural skills.

| | Mobile | | Desktop | | Web | | |
| --- | --- | --- | --- | --- | --- | --- | --- |
| **Agent Model** | **Text** | **Icon** | **Text** | **Icon** | **Text** | **Icon** | **Overall** |
| Operator (OpenAI, 2025) | 47.3 | 41.5 | 90.2 | 80.3 | 92.8 | 84.3 | 70.5 |
| Claude 3.7 Sonnet (Anthropic, 2025b) | - | - | - | - | - | - | 87.6 |
| UI-TARS-1.5 (Qin et al., 2025) | - | - | - | - | - | - | 94.2 |
| Seed-1.5-VL (Guo et al., 2025) | - | - | - | - | - | - | 95.2 |
| SeeClick (Cheng et al., 2024) | 78.4 | 50.7 | 70.1 | 29.3 | 55.2 | 32.5 | 55.1 |
| OmniParser-v2 (Yu et al., 2025) | 95.5 | 74.6 | 92.3 | 60.9 | 88.0 | 59.6 | 80.7 |
| Qwen2.5-VL-3B (Bai et al., 2025) | 93.4 | 73.5 | 88.1 | 58.6 | 88.0 | 71.4 | 80.9 |
| UI-TARS-2B (Qin et al., 2025) | 95.2 | 79.1 | 90.7 | 68.6 | 87.2 | 78.3 | 84.7 |
| OS-Atlas-Base-4B (Wu et al., 2024) | 95.2 | 75.8 | 90.7 | 63.6 | 90.6 | 77.3 | 85.1 |
| OS-Atlas-Base-7B (Wu et al., 2024) | 96.2 | 83.4 | 89.7 | 69.3 | 94.0 | 79.8 | 87.1 |
| JEDI-3B (Xie et al., 2025) | 96.6 | 81.5 | 96.9 | 78.6 | 88.5 | 83.7 | 88.6 |
| Qwen2.5-VL-7B (Bai et al., 2025) | 97.6 | 87.2 | 90.2 | 74.2 | 93.2 | 81.3 | 88.8 |
| UI-TARS-72B (Qin et al., 2025) | 94.8 | 86.3 | 91.2 | 87.9 | 91.5 | 87.7 | 90.3 |
| UI-TARS-7B (Qin et al., 2025) | 96.9 | 89.1 | 95.4 | 85.0 | 93.6 | 85.2 | 91.6 |
| JEDI-7B (Xie et al., 2025) | 96.9 | 87.2 | 95.9 | 87.9 | 94.4 | 84.2 | 91.7 |
| GUI-Owl-7B (Ye et al., 2025) | 99.0 | 92.4 | 96.9 | 85.0 | 93.6 | 85.2 | 92.8 |
| GUI-Owl-32B (Ye et al., 2025) | 98.6 | 90.0 | 97.9 | 87.8 | 94.4 | 86.7 | 93.2 |
| UI-Venus-Ground-7B (Gu et al., 2025) | 99.0 | 90.0 | 97.0 | 90.7 | 96.2 | 88.7 | 94.1 |
| UI-Venus-Ground-72B (Gu et al., 2025) | 99.7 | 93.8 | 95.9 | 90.0 | 96.2 | 92.6 | **95.3** |
| K²-Agent (Executor only) | 96.9 | 80.6 | 95.9 | 83.6 | 95.3 | 90.6 | 91.3 |

knowledge base ($K_G$), issues a clear and logical sub-goal (knowing what). The C-GRPO-trained executor then successfully grounds this sub-goal into a precise, low-level action on the screen (knowing how), seamlessly navigating between creating a note in one app and sharing its content via another.

### E.2.2 ANALYSIS OF GENERALIZATION AND FAILURE MODES

Our framework demonstrates powerful dual-generalization capabilities, where both high-level declarative knowledge and low-level procedural skills transfer effectively to unseen tasks.

**Declarative Knowledge Generalization.** We evaluated the planner's ability to adapt its declarative knowledge to novel tasks in the AitW benchmark. For each new task type, the SRLR loop was bootstrapped from a single demonstration, allowing the planner to rapidly form a new strategy. Figure 13 showcases the agent successfully completing two distinct, unseen tasks. This demonstrates that the SRLR self-evolution process produces robust, high-level strategies that are not mere scripts but generalizable plans that can be effectively applied to solve problems in new applications.

**Procedural Skill Generalization.** Figure 14 provides a visual testament to the executor's procedural generalization. Though trained only on AndroidWorld, the executor correctly identifies target UI elements on unseen mobile, desktop, and web interfaces from the ScreenSpot-v2 benchmark. This confirms that C-GRPO training develops a fundamental, platform-agnostic visual grounding capability rather than overfitting to specific app UIs.

**Failure Analysis.** Our analysis of failure cases reveals a consistent pattern: the majority of remaining errors do not stem from flaws in K²-Agent's framework (i.e., incorrect high-level plans or imprecise low-level actions), but from the intrinsic limitations of the current backbone VLM. Figure 12 visualizes a typical example from the `RetroCreatePlaylist` task, where the agent correctly identifies all songs but fails to add them in the strictly specified order. Other similar failures, such as those in `ExpenseAddMultipleFromMarkor` (requiring multi-document reasoning) and `MarkorMergeNotes` (requiring long-context text handling), also point to challenges in complex instruction following and long-horizon reasoning. These tasks represent the current frontier for foundation models, and we anticipate that as backbone models improve, these failures can be overcome without changes to the K²-Agent framework itself.

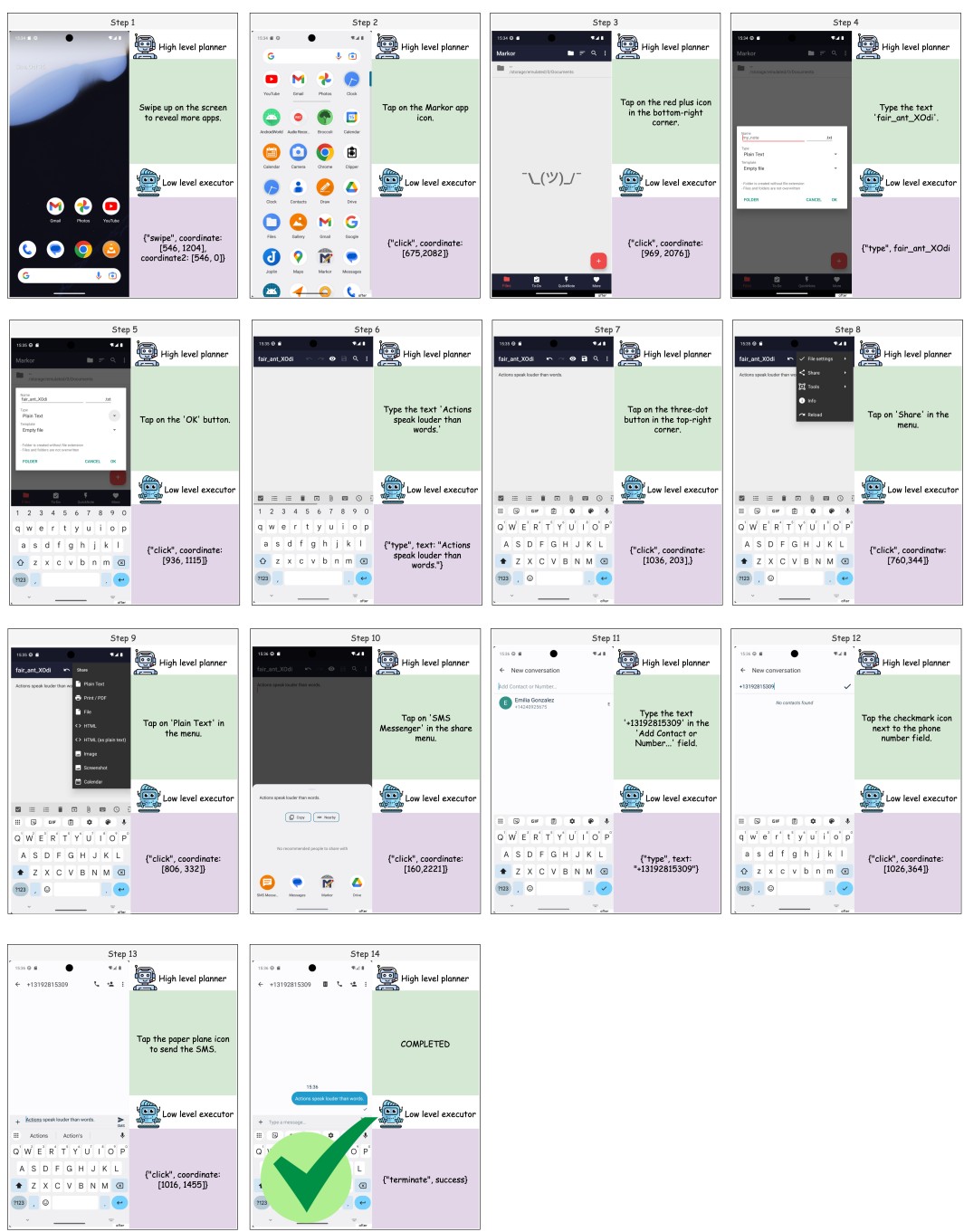

Figure 11: **Successful execution of the complex, multi-app task `MarkorCreateNoteAndSms`.**

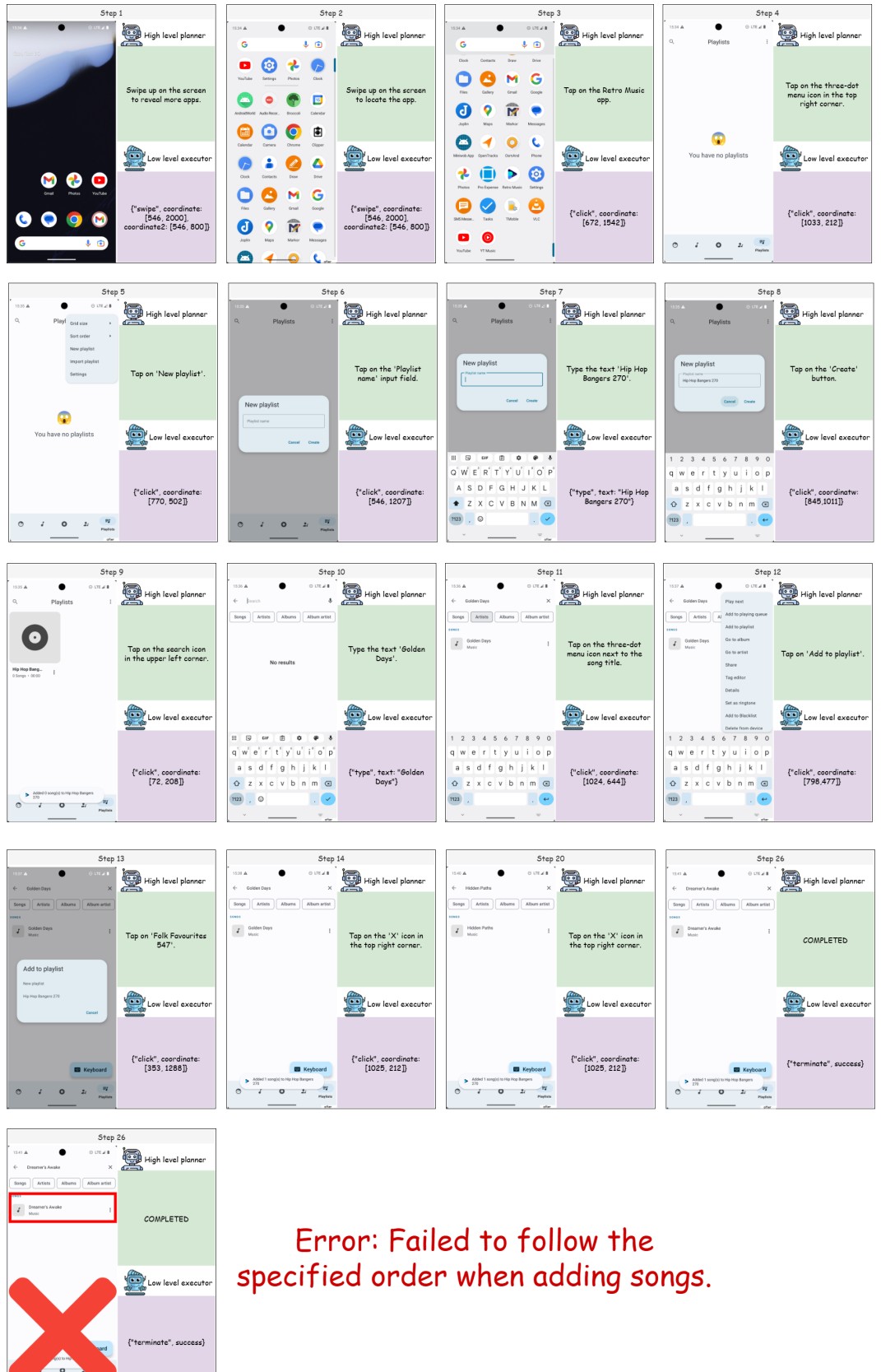

Figure 12: A representative failure case from the `RetroCreatePlaylist` task.

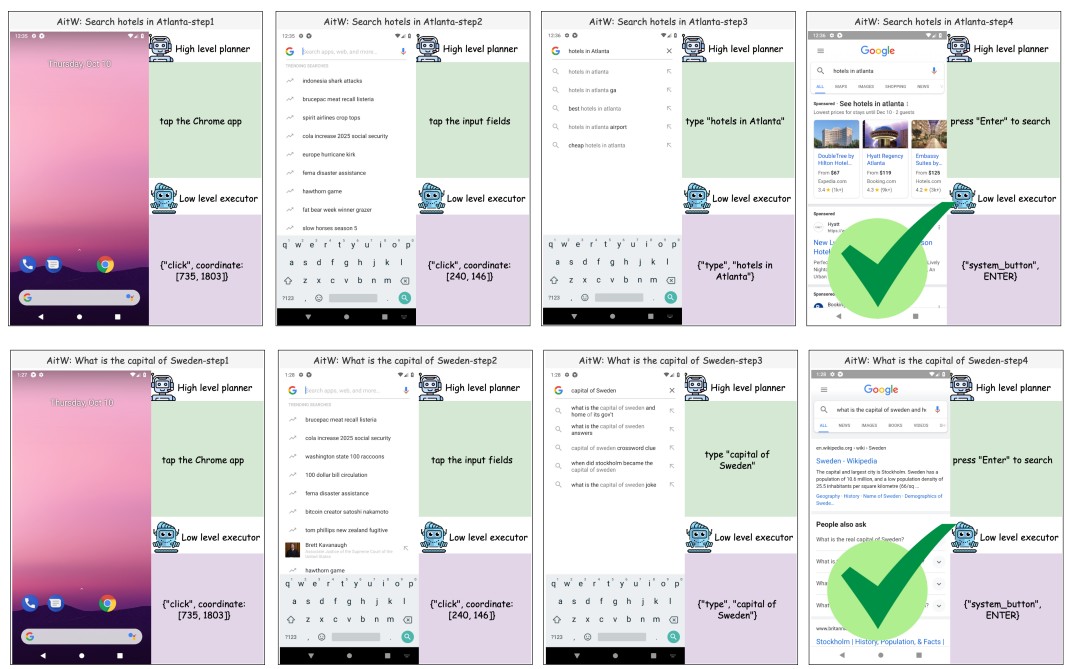

Figure 13: Declarative knowledge generalization on Android-in-the-Wild (AitW).

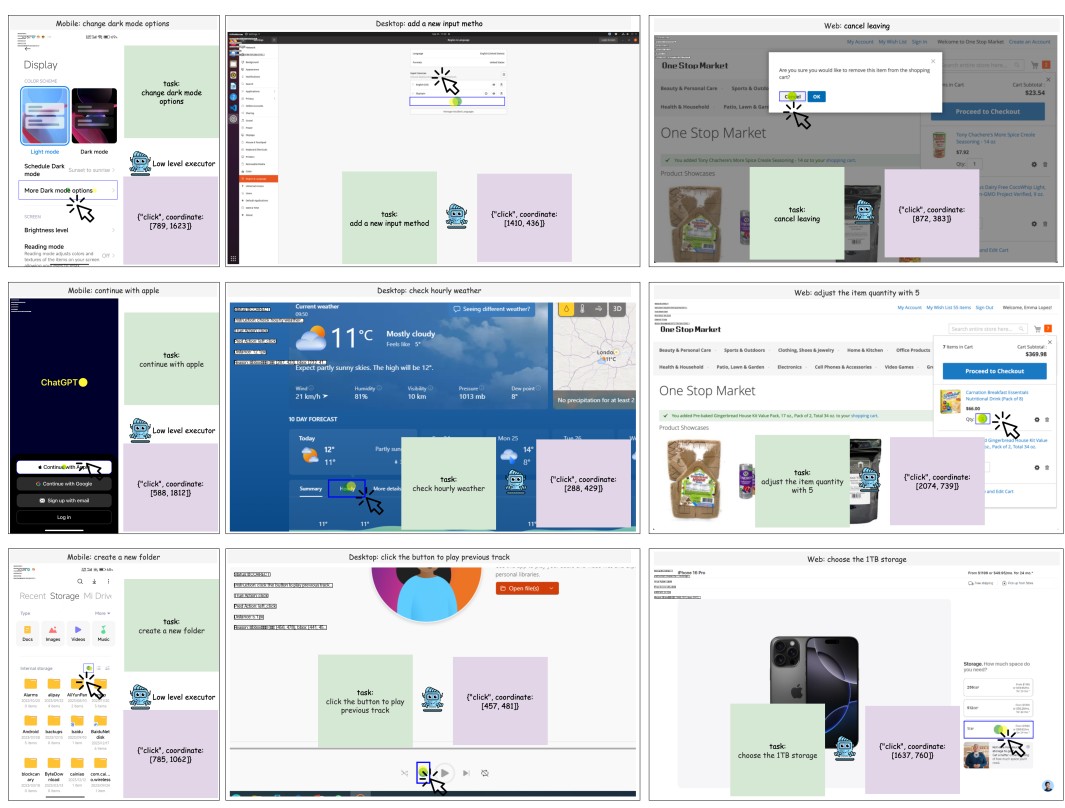

Figure 14: Zero-shot procedural skill transfer to the ScreenSpot-v2 benchmark.

