# OpenReview forum: "K²-Agent: Co-Evolving Know-What and Know-How for Hierarchical Mobile Device Control"
_ICLR.cc/2026/Conference — ICLR 2026 Poster_

### Official Review · Reviewer_UG52 · 2025-10-25

**Soundness:** 2
**Presentation:** 3
**Contribution:** 2
**Rating:** 6
**Confidence:** 1

**Summary:**

The paper **“K²-Agent: Co-Evolving Know-What and Know-How for Hierarchical Mobile Device Control”** introduces a cognitively inspired hierarchical agent that explicitly separates declarative (“know-what”) and procedural (“know-how”) knowledge to improve performance on mobile device control tasks. The system combines a **symbolic planner** that refines task knowledge through the **SRLR (Summarize–Reflect–Locate–Revise)** self-evolution loop, and a **reinforcement learning executor** trained with **C-GRPO (Curriculum-Guided Group Relative Policy Optimization)**. This dual-loop architecture enables co-evolution: the planner learns better strategies from execution feedback, while the executor gains structured guidance from the planner’s task decomposition. Experiments on the AndroidWorld benchmark demonstrate that K²-Agent achieves a **76.7% success rate**, surpassing prior open-source baselines while requiring minimal supervision (one demonstration per task). The work bridges symbolic reasoning and embodied learning in a cognitively grounded way, suggesting potential for broader applications in generalist AI systems.

**Strengths:**

- **Novel cognitive framework:** The clear distinction and co-evolution of declarative and procedural knowledge mirror human learning mechanisms, offering both conceptual clarity and practical benefits.
- **Methodological innovation:** The SRLR loop provides a systematic mechanism for reflective reasoning and self-correction, while C-GRPO elegantly handles data imbalance and sparse rewards through curriculum design.
- **Strong empirical results:** The system achieves state-of-the-art performance on AndroidWorld with extremely low data requirements, showing both efficiency and robustness.
- **Interpretable learning process:** The explicit task knowledge base and revision history improve transparency and allow fine-grained analysis of reasoning and execution errors.
- **Broader relevance:** The cognitive analogy to human dual-memory systems and the hierarchical design are relevant to multiple domains beyond mobile control, such as robotics and multimodal planning.

**Weaknesses:**

- **Limited evaluation diversity:** The experiments focus primarily on AndroidWorld; results on other benchmarks (e.g., WebArena, mobile navigation, or real-device tasks) would strengthen claims of generalization.
- **Scalability concerns:** The SRLR loop may involve significant overhead as task complexity grows, especially if symbolic reasoning steps become large or interdependent.
- **Ambiguity in error localization:** The “Locate” phase of SRLR appears manually defined or heuristic-based; it is unclear how reliably the system identifies and generalizes error points without human intervention.
- **Comparative analysis:** The work could benefit from deeper comparisons with alternative hybrid systems that combine reasoning and RL (e.g., ReAct, Voyager, or RPA-style architectures).
- **Reproducibility limitations:** Key implementation details—such as how knowledge base edits are represented or parameterized—are under-specified, making replication challenging for other researchers.

**Questions:**

1. How scalable is the SRLR loop when applied to tasks requiring multiple interdependent app contexts (e.g., cross-app workflows)?
2. Could C-GRPO be adapted for domains beyond visual mobile control—such as web navigation or embodied robotics—and if so, what adjustments would be required?
3. Does the declarative knowledge base support compositional reuse, allowing the agent to combine skills learned in separate tasks into new ones?
4. How sensitive is the system to the initial demonstration quality—can it recover from a poorly demonstrated “seed” trajectory?
5. How are symbolic edits validated to prevent catastrophic forgetting or the accumulation of contradictory task rules over long SRLR iterations?

---

> ### Author Response · Authors · 2025-11-24
> **General Response to Reviewer UG52 (0)**
>
> We sincerely thank the reviewer for the comprehensive and insightful feedback covering evaluation diversity, scalability, and reproducibility. In response, **we have uploaded a revised 10-page manuscript** that incorporates extensive new experiments and clarifications. Below is a summary of our major revisions:
>
> 1.  **Expanded Evaluation Scope (Response to Q1 & Q7):** We significantly broadened our evaluation beyond AndroidWorld. We added **ScreenSpot-v2** (Desktop/Web) and **Android-in-the-Wild (AitW)** benchmarks in the manuscript. As shown in the new **Table 2 & 3**, K²-Agent achieves strong zero-shot transfer performance across apps (86.5%) and platforms (91.3%), validating its generalization capability.
> 2.  **Clarified Scalability & Robustness (Response to Q2, Q6, Q9, Q10):** We explicitly analyzed the scalability of the SRLR loop for cross-app workflows and its robustness against imperfect demonstrations. We clarified that SRLR is a training-only process with linear complexity w.r.t. trajectory length, incurring zero inference overhead.
> 3.  **Enhanced Reproducibility (Response to Q3 & Q5):** To facilitate replication, we have provided the **full prompts** for all SRLR modules in **Appendix B.1.3**, clarified the atomic edit operations in **Sec. 4.2**, and uploaded the core codebase in the supplementary material.
>
> Finally, we have further enriched the manuscript with **rigorous ablation studies (Sec. 5.3) and hyperparameter sensitivity analyses (Sec. 5.4)**, ensuring a more comprehensive and empirically robust validation of our framework. We believe these updates thoroughly address your concerns and significantly strengthen the paper's contribution.

---

> ### Author Response · Authors · 2025-11-24
> **Response to Reviewer UG52 (1/4)**
>
> ### **Q1: Evaluation Diversity (Generalization to other benchmarks)**
>
> We thank the reviewer for emphasizing the importance of diverse evaluation. We agree that verifying generalization beyond a single benchmark is critical.
>
> While AndroidWorld serves as our primary controlled environment for rigorous training and ablation, we have explicitly expanded our evaluation scope in the revised manuscript (**Sec. 5.2**) to include **ScreenSpot-v2** and **Android-in-the-Wild (AitW)**. These benchmarks cover unseen apps, novel tasks, and entirely different operating systems (Desktop/Web).
>
> To rigorously quantify generalization, we summarize the overlap and zero-shot performance across three progressive levels in **Table R2** :
>
> **Table R2: Overlap quantification and zero-shot transfer performance.**
>
> | Evaluation Set | App Overlap | Episode Overlap | Generalization Level | Performance |
> | :--- | :--- | :---: | :--- | :---: |
> | **AndroidWorld (Test)** | 100% | 0% | Cross-instance | 76.1% SR |
> | **AitW (General subset)** | ≈36% | 0% | Cross-app | 86.5% SR |
> | **ScreenSpot-v2 (Desktop/Web)** | 0% | 0% | Cross-platform | 91.3% Acc |
>
> * **Cross-instance (AndroidWorld):** Training and testing use the same 20 apps but differ in instantiation parameters and seeds. The 0% episode overlap ensures the agent generalizes to dynamic content rather than memorizing trajectories. As shown in **Table 1 (Sec. 5.1)**, K²-Agent achieves **76.1% SR** under this setting.
> * **Cross-app (AitW):** We evaluate the executor zero-shot on AitW. Among the 11 apps, 7 are entirely unseen during training (e.g., YouTube, Maps), and all task templates are novel. The **86.5% success rate** proves the agent learned transferable interaction patterns robust to novel UI layouts (see **Table 2 in Sec. 5.2**).
> * **Cross-platform (ScreenSpot-v2):** We further transfer the AndroidWorld-trained executor to the Desktop (Windows/macOS) and Web splits of ScreenSpot-v2. Despite the complete domain shift (0% overlap in apps/platforms), the executor achieves **91.3% grounding accuracy**, performing competitively with specialized agents trained on massive multi-platform datasets (see **Table 3 in Sec. 5.2**).
>
> These results demonstrate that K²-Agent is not limited to the training distribution of AndroidWorld but exhibits strong robustness across unseen apps and platforms.
>
> ### **Q2: Scalability of the SRLR Loop**
>
> We appreciate the concern regarding scalability. In our design, the SRLR loop is used **only during training** to refine the knowledge base ($K_G$); at test time, the planner simply queries the finalized rules to output one sub-goal per step, so there is **no SRLR overhead at inference**.
>
> Each SRLR iteration operates on a **single trajectory** and performs local atomic edits on a small subset of rules. We cap the number of iterations per task (**up to 10** in our implementation). Empirically, we observe that the knowledge base typically stabilizes within a few iterations (**often within 3–5**), whereas the executor requires thousands of RL rollouts. Thus, the total wall-clock cost of SRLR remains negligible compared to RL training.
>
> ### **Q3: Ambiguity in Error Localization (SRLR)**
>
> Thanks for pointing this out. We clarify that the Locate phase is **fully automated** and involves **no human intervention**. It is directly driven by the VLM-based Reflect module. Concretely, Reflect predicts the failing step (e.g., “at step 4 the agent clicked ‘Cancel’ instead of ‘Save’”), and Locate deterministically maps this step index to the corresponding rule in $K_G$ and marks only that rule for revision—no human intervention is involved. The Revise phase then generalizes this rule (e.g., from “bottom-right button” to “the ‘Save’ button”), so localization happens at the rule level and the fix transfers to future episodes.
>
> We have provided the **exact prompts** used for the Locate phase in **Appendix B.1.3** and added a concrete **evolution case study** in **Appendix B.1.5** to demonstrate how the module automatically identifies and fixes errors without human intervention.

---

> ### Author Response · Authors · 2025-11-24
> **Response to Reviewer UG52 (2/4)**
>
> ### **Q4: Comparative Analysis with Hybrid Systems**
>
> We thank the reviewer for this insightful suggestion to position our work against broader hybrid architectures. We have added a comprehensive discussion in **Appendix C** of the revised manuscript.
>
> Below is a summary of the key distinctions we elaborated on:
>
> 1.  **vs. ReAct-style (Interleaved Reasoning & Acting):**
>     Our **"No Hierarchy" baseline** (Table 4) essentially represents a ReAct-like setup where a monolithic policy handles both reasoning and acting from raw screenshots. This variant achieves only **35.3% success**, whereas the full K²-Agent reaches **76.1%**. This empirical gap demonstrates that explicitly decoupling "know-what" (SRLR planner) from "know-how" (RL executor) is critical for preventing the cognitive drift and context overflow often seen in ReAct-style policies on long-horizon GUI tasks.
>
> 2.  **vs. Voyager-style (Code-based Skill Libraries):**
>     Systems like Voyager evolve skills as executable code (e.g., Python functions) relying on structured APIs (like Mineflayer). In mobile GUI control based on **raw screenshots**, such stable APIs are unavailable. K²-Agent instead evolves skills as **parametric policies** (neural network weights) trained via C-GRPO. This allows our method to operate in pixel-based environments where code-generation approaches are inapplicable.
>
> 3.  **vs. RPA-style (Rule-based Automation):**
>     Classical Robotic Process Automation (RPA) relies on static scripts and hand-crafted rules, which are brittle to UI updates or layout changes. In contrast, K²-Agent is **data-driven and adaptive**: it iteratively revises its symbolic knowledge via SRLR and trains a generalist executor that transfers zero-shot to unseen apps (AitW) and platforms (ScreenSpot-v2), offering a robust alternative to fixed RPA pipelines.
>
> ### **Q5: Reproducibility of $K_G$ Edits and SRLR Implementation**
>
> We appreciate the concern about reproducibility. The concrete implementation details are specified as follows:
> - **How $K_G$ is represented and edited**
>
> KG is implemented as a line-structured natural-language rule list, and edits are atomic line-level add / delete / modify operations. Appendix Sec. “Implementation Details of SRLR Modules” and “Evolution of the Knowledge Base ” provide full before–after examples ($K_G^0$→$K_G^1$→$K_G^2$) for a real task, making the edit process explicit.
> - **SRLR module specification**
>
> The exact prompts and output formats for Summarize, Reflect, Locate, and Revise are given verbatim in **Appendix Sec. “Implementation Details of SRLR Modules”**, so each stage can be directly re-implemented.
>
> The core SRLR and C-GRPO implementation is included in the supplementary material, and we are preparing a public release with full configs, models and checkpoints to further support independent replication.
>
> ### **Q6: Scalability of SRLR for Cross-App Workflows**
>
> The SRLR loop scales naturally to cross-app workflows because it operates at the **task level**, not the app level.
> 1. **Task-level abstraction, linear growth with steps**
>
> The knowledge base $K_G$ encodes an ordered sequence of subgoals for an entire task (e.g., “open the gallery and read the receipt → switch to Markor → create a markdown note with the required format”), irrespective of how many apps are involved. SRLR (Summarize → Reflect → Locate → Revise) always runs over a single task trajectory, so its cost grows roughly linearly with the number of steps in that trajectory, rather than with the number of apps or contexts.
>
> 2. **Already exercised on cross-app tasks in AndroidWorld**
>
> AndroidWorld itself contains multi-app tasks with interdependent contexts (e.g., **MarkorTranscribeReceipt**, where the agent must remember content from an image app and then write it into Markor under a specific template). We apply SRLR to these cross-app tasks without any change to the algorithm and observe no degradation compared to single-app tasks.
>
> Overall, SRLR treats cross-app workflows as longer single tasks with more steps, and both the design and our empirical results indicate that no additional mechanism is required for it to remain scalable in such settings.

---

> ### Author Response · Authors · 2025-11-24
> **Response to Reviewer UG52 (3/4)**
>
> ### **Q7: Applicability of C-GRPO Beyond Mobile GUIs**
>
> We appreciate the reviewer’s forward-looking question. Fundamentally, C-GRPO is **domain-agnostic**: it relies only on perceptual observations, sparse rewards, and identifiable error modes.
>
> **(1) Proven Generalization to Web and Desktop**
> We have explicitly validated the adaptability of C-GRPO to web and desktop domains without any algorithmic modification. As shown in **Table R3** (full version in sec5.2), the executor trained *exclusively* on AndroidWorld transfers zero-shot to the **ScreenSpot-v2** benchmark, achieving **95.3% accuracy on Web** and **95.9% on Desktop (Text)**.
>
> **Table R3: Zero-shot performance on ScreenSpot-v2 (Transfer from AndroidWorld).**
>
> | Agent Model | Mobile Text | Mobile Icon | Desktop Text | Desktop Icon | Web Text | Web Icon | **Overall** |
> | :--- | :---: | :---: | :---: | :---: | :---: | :---: | :---: |
> | Operator (OpenAI) | 47.3 | 41.5 | 90.2 | 80.3 | 92.8 | 84.3 | 70.5 |
> | UI-TARS-72B | 94.8 | 86.3 | 91.2 | 87.9 | 91.5 | 87.7 | 90.3 |
> | GUI-Owl-32B | 98.6 | 90.0 | 97.9 | 87.8 | 94.4 | 86.7 | 93.2 |
> | **K²-Agent** | **96.9** | **80.6** | **95.9** | **83.6** | **95.3** | **90.6** | **91.3** |
>
> This result confirms that the procedural skills learned via C-GRPO are not limited to mobile interfaces but generalize to broader GUI domains.
>
> **(2) Potential Adaptation to Embodied Robotics**.
> The framework adapts naturally by abstracting "actions" from UI primitives to high-level robot skills. For instance, pixel coordinates map to continuous 6-DoF poses, and atomic actions map to primitives like grasp(object) or move_to(pose). Consequently, our core mechanisms—error-decoupled replay (balancing grasp failures vs. motion errors) and curriculum via partial demonstrations (annealing expert waypoints)—carry over directly.
>
> ### **Q8: Compositional Reuse of Declarative Knowledge**
>
> Yes, the declarative knowledge base ($K_G$) supports compositional reuse through its **modular structure** and **hierarchical decomposition**, which we characterize in two aspects:
>
> **(1) Structural Compositionality (Representation Level)**
> The $K_G$ consists of explicit, modular natural-language rules. This representation is inherently compositional: complex tasks are represented not as monolithic scripts, but as ordered sequences of independent logical units (sub-goals). In composite tasks like `MarkorCreateNoteAndSms`, we observe that the SRLR loop evolves a $K_G$ that effectively concatenates the logical blocks of simpler tasks (e.g., "Create Note" rules + "Send SMS" rules).
>
> **(2) Procedural Composition (Execution Level)**
> The decomposition provided by $K_G$ enables the **Low-Level Executor** to achieve massive compositional reuse. Because $K_G$ breaks a new, complex task into standard sub-goals (e.g., "Tap the Save button"), the executor can directly reuse its learned atomic skills (grounding and execution) acquired from completely different tasks. This ensures that skills learned in Task A (e.g., operating a specific UI widget) are immediately available for any new Task B that requires the same interaction, requiring no re-training of the procedural policy.

---

> ### Author Response · Authors · 2025-11-24
> **Response to Reviewer UG52 (4/4)**
>
> ### **Q9: Sensitivity to Initial Demonstration Quality**
>
> We clarify that while our method benefits from a functional seed demonstration, it is **not brittle to imperfections**. The system employs a dual-mechanism defense to recover from suboptimal demos:
>
> **(1) Summarization at Initialization**
> During the `Summarize` stage, the planner actively abstracts away instance-specific noise (e.g., specific filenames, random scrolls, or contact names). Instead of hard-wiring these literals, it extracts high-level structural rules and parameter constraints. This filters out "parameter noise" immediately.
>
> **(2) Correction via Evolution**
> If the demonstration contains logic flaws (e.g., missing a precondition step), the `SRLR` loop acts as a self-correction mechanism.  When the initial rule leads to a runtime failure, *Reflect* detects the mismatch, *Locate* finds the flawed rule, and *Revise* updates it based on the new environmental feedback. We observed this behavior in the `AudioRecorder` task (detailed in **Appendix B.1.5**). The initial seed demonstration omitted a critical step to clear the input field. Consequently, the initial plan $K_{G}^0$ failed. However, the SRLR loop successfully identified this cause and inserted the missing "long-press backspace" step into $K_{G}^2$.
>
> This confirms that the system can successfully recover from partially flawed or incomplete seed trajectories, provided the overall task intent is discernible.
>
> ### **Q10: Validation of Symbolic Edits and Avoiding Contradictory Rules**
>
> We agree that unconstrained symbolic edits could theoretically lead to inconsistencies, but in K²-Agent, SRLR controls this through strict evidence grounding and iterative environmental validation.
>
> First, edits to $K_G$ are never free-form. In the **Locate** phase, the model identifies the specific lines in the current prompt responsible for the failure; in the **Revise** phase (see prompts in **Appendix B.1.3**), it is explicitly instructed to modify only those targeted lines while keeping them consistent with the original seed demonstration. Crucially, new rules **overwrite** old ones rather than being appended in parallel, ensuring that contradictory instructions do not accumulate over time.
>
> Second, every updated $K_G$ is immediately validated through execution. If a revision introduces a new systematic error (regression), the resulting failure triggers a subsequent SRLR cycle to further correct the offending rule. The **AudioRecorder example (Appendix B.1.5)** illustrates this self-correcting loop. Finally, since SRLR runs for a capped number of iterations (max 10) on a small set of rules, the combination of atomic overwrites, environment-based validation, and bounded updates makes catastrophic forgetting or long-term rule conflicts unlikely in practice.

---

> > ### Comment · Reviewer_UG52 · 2025-11-24
> > **Response to rebuttal**
> >
> > We appreciate the authors' detailed and comprehensive rebuttal to the concerns raised, particularly those related to generalization, complexity, and implementation transparency.
> >
> > The reviewer acknowledges that the extensive revisions and clarifications—including the new experimental results and the added appendices—have significantly strengthened the paper's empirical foundation and technical clarity.
> >
> > ### **Summary of Key Clarifications Acknowledged:**
> >
> > * **Expanded Evaluation (Q1, Q7):** The inclusion of zero-shot transfer performance on **ScreenSpot-v2** (cross-platform, 91.3% accuracy) and **Android-in-the-Wild (AitW)** (cross-app, 86.5% success rate) robustly validates the dual generalization claim.
> > * **SRLR Mechanism (Q3, Q9, Q10):** The clarification that the SRLR process is fully automated, human-free, and capable of recovering from sub-optimal initial demonstrations confirms the self-correcting nature of the declarative knowledge loop.
> > * **Implementation & Scalability (Q2, Q5, Q6):** The clear statement that the SRLR loop runs exclusively **during training** with **zero inference overhead** addresses the scalability concerns. The provision of verbatim prompts and explicit atomic edit rules enhances reproducibility.
> >
> > ### **Final Decision**
> >
> > While the reviewer acknowledges the technical strengths and significant improvements made in the revised manuscript, particularly the innovative **knowledge decoupling paradigm** and the strong SOTA results on AndroidWorld, the final score will remain unchanged.
> >
> > The reviewer, who specializes outside of the core agent field, feels the paper's overall contribution is now clearly defined and strongly supported. We believe the paper's final fate should be determined by the Area Chair, who can integrate the robust evidence provided in this rebuttal with the specialized feedback from the other reviewers.
> >
> > **Recommendation to AC:** The reviewer recommends that the Area Chair carefully consider the substantial empirical improvements (new generalized SOTA) and the detailed technical clarifications when making the final decision.
> >
> > We thank the authors again for their diligence and look forward to seeing the final version of this work.

---

### Official Review · Reviewer_tNhe · 2025-10-29

**Soundness:** 3
**Presentation:** 2
**Contribution:** 2
**Rating:** 4
**Confidence:** 3

**Summary:**

This work tackles long‑horizon mobile device control and argues that agents should separate two kinds of knowledge for accurate planning and execution: (i) declarative “knowing what” for task planning, and (ii) procedural “knowing how” for precise UI actions. The proposed K²‑Agent is a two‑layer planner–executor:

High‑level planner (know‑what): a training‑free VLM that bootstraps from one demonstration per task and maintains a textual task knowledge base (KG) updated by a Summarize Reflect Locate Revise (SRLR) loop. SRLR distills an initial plan from a demo, analyzes failures, identifies the wrong decision points, and applies atomic edits to the knowledge base.

Low‑level executor (know‑how): a trainable VLM optimized with Curriculum‑guided GRPO (C‑GRPO). C‑GRPO (1) builds error‑decoupled replay pools for type vs. parameter errors, and (2) uses dynamic demonstration injection to prepend variable‑length expert prefixes for difficult samples; the policy is updated with GRPO.

**Strengths:**

* The experimental results are strong. Especially, the KG graph being transferred to different VLMs, and the low-level executor being transferred to AiTW and ScreenSpot-v2 is interesting.
* Proposing to train the low-level executor with reinforcement learning while refining the planner’s knowledge base through training-free self-evolution (SRLR) offers an efficient division between low-level actions and declarative knowledge planning & refinement.

**Weaknesses:**

* The overall flow of section 4 itself was easy to understand, but specifically what each component is and how they operate is unclear in several parts. Specifically, (1) what 'atomic' edits are and how they are applied to the KG was unclear until looking at the appendix. (2) In dynamic demonstration injection, what 'variable length expert prefix' is unclear. (3) What the single demonstration that is provided is, and whether this is identical or different from the AndroidWorld benchmark is unclear
* If the benchmark provides single expert demonstrations per task category, this may be providing more information than other results

**Questions:**

* What is the motivation of applying the dynamic length prefix? Why does this improve performance?
* What is the dynamic length expert prefix and how is the length dynamically applied? For example,  if the expert prefix something like 'Be sure to answer in the format of ...', how do you dynamically change the length of this prefix?
* Regarding the single demonstration provided, is the provided task excluded from the benchmark? Also, if a successful task is provided per category of the AndroidWorld benchmark, it seems not fair to say that this method utilizes less information than other methods that only use screenshots or A11y trees.
* How is the KG constructed from the high-level planner utilized by the low-level executor? The reverse is clear to me, the demonstrations of the low-level executor fails is provided to the planner.

---

> ### Author Response · Authors · 2025-11-24
> **General  Response to Reviewer tNhe (0)**
>
> We sincerely thank the reviewer for the insightful questions regarding the motivation, mechanism, and fairness of our approach. In response, **we have uploaded a revised 10-page manuscript** containing new definitions, rigorous ablation studies, and extended clarifications. Below is a summary of our major revisions:
>
> 1.  **Clarified Dynamic Demonstration Injection (Response to Q2 & Q3):** We significantly expanded **Sec. 4.3** to define the expert prefix as "atomic action steps" (not text hints) and clarified the dynamic length scheduling mechanism. To empirically validate this design, we added a **Parameter Sensitivity Analysis (Sec. 5.4, Fig. 6)**, demonstrating that the temperature parameter $T$ effectively controls the curriculum and is essential for convergence.
> 2.  **Validated Fairness & Generalization (Response to Q4):** We added a detailed **Data Leakage Analysis (Appendix D, Table R2)**. We explicitly quantify the overlap between training and testing, and report **zero-shot transfer results** on unseen apps (AitW) and platforms (ScreenSpot-v2) to prove that our single-demonstration approach learns robust skills rather than memorizing benchmarks.
> 3.  **Explicit Definitions for Edits & KG (Response to Q1 & Q5):** We moved the definitions of atomic edits to the main text (**Sec. 4.2**) and clarified the specific two-stage information flow from $K_G$ to the Executor (**Sec. 4.1**), ensuring the system architecture is transparent.
>
> We believe these updates directly address your concerns and strengthen the paper's clarity and rigor.

---

> ### Author Response · Authors · 2025-11-24
> **Response to Reviewer tNhe (1/3)**
>
> ### **Q1: Clarification on Atomic Edits for $K_G$**
>
> We apologize that these definitions were initially buried in the Appendix. To improve clarity, we have **revised Section 4.2** in the main text to explicitly define the four atomic edit operators used by the planner:
>
> 1.  **ADD:** Insert a missing logical step or precondition (e.g., "Check if the list is empty first").
> 2.  **DELETE:** Remove a redundant, erroneous, or hallucinated step.
> 3.  **UPDATE:** Modify the description or parameters of an existing step (e.g., changing specific text content).
> 4.  **HIGHLIGHT:** Emphasize critical constraints or attention points (e.g., "MUST wait for the page to load").
>
> The planner outputs these structured tool calls to perform local surgeries on the knowledge base, rather than rewriting the entire text from scratch. This ensures the evolution is stable and incremental.
>
> ### **Q2: Motivation and Mechanism of Dynamic Demonstration Injection**
>
> We thank the reviewer for this constructive question. In the revised manuscript, we have clarified the intuition in **Sec. 4.3** and added explicit ablation studies in **Sec. 5.3 (Fig. 5b)** to empirically validate this design.
>
> **(1) Motivation: Overcoming Exploration Hurdles in Large Action Spaces**
> The action space in mobile GUIs is vast (combining screen coordinates and text inputs), and rewards are extremely sparse (often only given at the end of a long trajectory). A policy starting from scratch faces an immense exploration difficulty; the probability of randomly stumbling upon a successful multi-step trajectory is near zero, meaning the model receives almost no positive learning signal to begin optimization. Our Dynamic Demonstration Injection serves as a curriculum. By prepending expert prefixes, we drastically shrink the effective search space and guarantee that the agent is exposed to successful states early in training. This bootstraps the learning process, ensuring the optimizer has positive gradients to work with from the start.
>
> **(2) Why it Improves Performance: Curriculum-Driven Internalization**
> Initially, the prefixes act as strong guidance (hints). As training progresses, we anneal the prefix length toward zero based on the sample difficulty and training stage. To maintain high rewards as the expert guidance vanishes, the policy is forced to internalize the skills into its own parameters, learning to rely solely on the original query and visual context. We validated this in our new ablation study under a strictly controlled compute budget. As shown in the **Green Curve in Figure 5(b)**, removing Dynamic Demonstration Injection causes the policy to suffer from severe oscillations and low rewards, as it fails to effectively explore the environment. In contrast, the full method (Blue Curve) converges stably, proving that this mechanism is essential for unlocking high performance.

---

> ### Author Response · Authors · 2025-11-24
> **Response to Reviewer tNhe (2/3)**
>
> ### **Q3: Definition and Scheduling of the Dynamic-Length Expert Prefix**
>
> We appreciate the opportunity to clarify the implementation details.
>
> (1) **What is dynamic length expert prefix?**
> First, we clarify that the "expert prefix" is defined over **atomic action steps** from the demonstration trajectory, **not** over partial natural-language instructions.
> * For each task, the expert demonstration is a sequence of $N$ complete low-level actions: $\tau_{\text{demo}} = [a_1, a_2, \dots, a_N]$.
> * The injected prefix is strictly a subset of these complete actions: $\tau_{\text{prefix}} = [a_1, \dots, a_m]$.
> * We **never** truncate inside a sentence or a function call; the prefix always consists of an integer number of executable steps to ensure valid context.
>
> **(2) How is the Length Applied Dynamically?**
> We determine the number of injected steps $m$ using the schedule defined in **Eq. 6** of the manuscript. The injected length $l$ is computed as:
> $$
> l = L \cdot \sigma(k) \cdot f_{\text{gate}}(d_i),
> $$
> where:
> * $L$ represents the total length (aligned with the total steps $N$ of the demonstration).
> * $\sigma(k) = \max(0, 1 - k / K_{\max})$ is the **linear annealing scheduler**, which monotonically decays as training step $k$ increases.
> * $f_{\text{gate}}(d_i) = \tanh(d_i / T)$ is the **difficulty gating function**, which assigns larger weights to harder samples (higher error rate $d_i$).
>
> In practice, we convert this continuous length $l$ into a discrete number of steps: $m = \lfloor l \rfloor$.
> * **Early in training / Hard samples:** $\sigma(k) \approx 1$ and $f_{\text{gate}}(d_i)$ is high. Thus $m \approx N$, and the model receives nearly the full demonstration as guidance.
> * **Late in training / Easy samples:** $\sigma(k) \to 0$ or $f_{\text{gate}}(d_i)$ drops. Thus $m \to 0$, and the prefix vanishes, forcing the model to rely solely on the query-only input.
>
> **(3) Empirical Verification via Temperature $T$ (New Sec. 5.4)**
> The temperature parameter $T$ in the gating function $f_{\text{gate}}$ is the key knob controlling this dynamic mechanism. To prove its effect, we added a sensitivity analysis in **Sec. 5.4 (Fig. 6)**:
> * **High $T$ ($T=5.0$):** The gating term $f_{\text{gate}}(d_i) \approx 0$, meaning the dynamic prefix is suppressed. The result (failure to converge) confirms that without this dynamic help, the task is too hard.
> * **Appropriate $T$ ($T=0.5$):** This setting activates the dynamic schedule described above, enabling the curriculum that leads to SOTA performance.
>
> We have revised **Section 4.3** to explicitly state that the prefix length refers to the "number of atomic expert actions" to avoid ambiguity. Additionally, the parameter sensitivity analysis regarding $T$ has been included in **Section 5.4** and **Figure 6**.

---

> ### Author Response · Authors · 2025-11-24
> **Response to Reviewer tNhe (3/3)**
>
> ### **Q4: Single Demonstration Usage: Benchmark Exclusion and Fairness**
>
> We thank the reviewer for this opportunity to clarify our experimental rigor and the context of our contributions.
>
> **(1) We use strict instance-level exclusion**
> We follow the official AndroidWorld protocol and maintain a strict separation between demonstration data and evaluation episodes. For each training task template, we use one successful trajectory from the training split to bootstrap SRLR. Test episodes are generated with different random seeds, so their parameters (e.g., contact names, filenames, settings values) differ from those in the demonstrations. No evaluation episode is ever used as a demonstration.
>
> **(2) Fairness and the "Less Information" Claim**
> We argue that our comparison is fair and, in fact, highlights our core advantage of Data Efficiency.
> It is common for learning-based GUI agents to pre-train on massive datasets (e.g., UI-Venus uses \~350k samples; MobileRL uses \~3k (2k AndroidWorld + 1k AndroidLab tasks)). These datasets inevitably cover the popular apps in AndroidWorld (Chrome, Settings, etc.). Compared to these baselines, K²-Agent achieves SOTA performance using orders of magnitude less supervision (~100 single demos vs. hundreds of thousands).
> And the fairness of a comparison typically relies on test-time constraints. At inference time, K²-Agent operates under the exact same constraints as standard screenshot-based baselines. It receives only the current screenshot and instruction, without accessing the demonstration or A11y tree.
>
> **(3) Generalization beyond the training distribution.**
> To definitively rule out the concern that performance stems from memorizing the single demonstration or overfitting to AndroidWorld's specific UI layouts, we explicitly quantify overlap and report zero-shot transfer at three levels (Table R2):
>
> Table R2: Overlap quantification and zero-shot transfer performance.
>
> | Evaluation Set | App Overlap | Episode Overlap | Generalization Level | Performance |
> | :--- | :--- | :---: | :--- | :---: |
> | AndroidWorld (Test) | 100% (same 20 apps) | 0% | Cross-instance | 76.1% SR |
> | AitW (General subset) | ≈36% (4/11 overlap) | 0% | Cross-app | 86.5% SR |
> | ScreenSpot-v2 (Desktop/Web) | 0% (new platforms) | 0% | Cross-platform | 91.3% Acc |
>
> * **Cross-instance (AndroidWorld):** Training and testing differ in instantiation parameters and seeds. The 0% episode overlap ensures the agent generalizes to dynamic content.
> * **Cross-app (AitW):** We evaluate the executor zero-shot on AitW. Among the 11 apps, 7 are entirely unseen (e.g., YouTube, Maps), and all task templates are novel (**0% template overlap**). The 86.5% success rate proves the agent learned transferable interaction patterns, not just AndroidWorld layouts.
> * **Cross-platform (ScreenSpot-v2):** We transfer the executor to Desktop/Web platforms with **0% overlap** in apps and templates. The 91.3% grounding accuracy confirms the procedural skills are robust to complete domain shifts.
>
>
> In summary, K²-Agent demonstrates that high performance can be achieved through efficient learning (via SRLR and C-GRPO) rather than massive data ingestion or test-time leakage. The strict train-test separation, combined with strong zero-shot transfer results, validates the robustness of our method.
>
> **Q5: Utilization of $K_G$ by the Low-Level Executor**
>
> We thank the reviewer for this question. We clarify that the knowledge base ($K_G$) is utilized **indirectly** by the low-level executor through the generation of sub-goals. The information flow operates as a two-stage cascade:
>
> 1.  **$K_G \rightarrow$ Planner ($\pi_H$) $\rightarrow$ Sub-goal ($z_t$):**
>     The high-level planner $\pi_H$ consults the declarative rules in $K_G$ to decompose the complex task into a specific, immediate natural-language sub-goal $z_t$ (e.g., "Tap the 'Add' button").
>
> 2.  **Sub-goal ($z_t$) $\rightarrow$ Executor ($\pi_L$) $\rightarrow$ Action:**
>     The low-level executor $\pi_L$ receives this sub-goal $z_t$ (paired with the current screenshot) as its instruction. It does **not** see $K_G$ directly.
>
> As the SRLR loop refines $K_G$, the planner generates more precise and logically sound sub-goals ($z_t$). This improved instruction makes the grounding task easier for the executor, improving performance without requiring architectural changes to the low-level model.

---

> ### Comment · Area_Chair_VAXa · 2025-11-25
> **Please participate in discussions with authors and other reviewers asap**
>
> Please kindly and actively participate in the review-author discussion if you haven't already, raise your further concerns so that the authors can explain more, and make your final decisions.
> Best, AC

---

> ### Comment · Reviewer_tNhe · 2025-11-28
>
> Thank you for clarifying several questions.
> In accordance to my questions being answered and the flow of the paper being better, I am willing to raise my score to 6, but the system does not allow it at the moment.
> Will update it after the issue is resolved.

---

### Official Review · Reviewer_2B17 · 2025-11-01

**Soundness:** 2
**Presentation:** 3
**Contribution:** 3
**Rating:** 4
**Confidence:** 2

**Summary:**

The paper proposes $K^2$-Agent, a hierarchical framework for mobile device control that explicitly decouples declarative “know-what” (a high-level planner distilled via a Summarize–Reflect–Locate–Revise, SRLR, loop) from procedural “know-how” (a low-level executor trained with a curriculum-guided C-GRPO objective). On AndroidWorld (116 tasks across 20 apps), the authors claim state-of-the-art 76.7% success “ranking 1st … among methods using only raw screenshots and open-source backbones” and show transfer to ScreenSpot-v2 and Android-in-the-Wild (AitW) without fine-tuning. The intended scope is screenshot-only agents on Android emulator tasks, plus cross-benchmark skill transfer.

**Strengths:**

* Clear hierarchical decomposition (SRLR + C-GRPO) with reasonable mathematical formalization (PPO-style clipping; explicit content/format rewards).
* The paper’s split between a high-level SRLR planner (“knowing what”) and a low-level C-GRPO executor (“knowing how”) isn’t just conceptual; you show the actual SRLR prompts and output formats. This makes the approach inspectable and reuse-friendly.
* Even with screenshot-only I/O, the result on AndroidWorld is pretty strong as shown in Table 1.
* The executor’s reward is specified precisely (format + content, Eq. (5) and total reward definition) and the action space is enumerated (Table 6), which helps reproduction and critique.

**Weaknesses:**

* The authors mention that “Human experts achieve about 80% average success” on AndroidWorld without details (line 373): annotator count, expertise, task sampling, time limits, or inter-rater protocol.
* The AndroidWorld results do not report of seeds, dispersion, or confidence intervals (CIs). Without multi-seed evaluation, the “SOTA” claim is statistically fragile.
* The executor is trained on 606 single-step samples derived from 116 demos (Appendix B.2.2). Even with the stated seed-based split controls, the paper doesn’t quantify app- or template-level overlap between training demonstrations and evaluation task distributions; leakage through UI idiosyncrasies remains plausible without stricter controls or OOD app tests.

**Questions:**

* How many seeds were used for Table 1 and Figure 1? Please report mean ± std over ≥3 seeds and add 95% CIs to plots.
* Right now, it is hard to tell whether the gains come from C-GRPO itself or. from extra compute or curriculum variations. You should compare GRPO vs C-GRPO with everything else identical: model init, data, rollout length, batch size, optimizer, KL/clip settings, and exactly the same number of environment interactions (and similar wall-clock within ±5%). Also please report the learning curves of these runs.
* Vary the key curriculum knob(s) you introduce (e.g., mixing ratios, scheduling temperature T if applicable) across a small grid (e.g., 3–5 values). For which hyperparameters is your method brittle or robust?

---

> ### Author Response · Authors · 2025-11-24
> **General Response to Reviewer 2B17 (0)**
>
> We sincerely thank the reviewer for the constructive criticism, particularly regarding statistical rigor and ablation controls. In response, **we have uploaded a revised 10-page manuscript** that incorporates new multi-seed experiments, controlled ablation studies, and sensitivity analyses to directly address your concerns. Below we summarize the main changes:
>
> 1.  **Statistical Robustness (Response to Q2):** We re-evaluated K²-Agent with **3 independent random seeds** on AndroidWorld. We updated **Table 1** (reporting mean $\pm$ std) and added **95% Confidence Intervals** to **Figure 1 & 5**, confirming our SOTA claim is statistically significant even with the lowest performing seed.
> 2.  **Strictly Controlled Ablation (Response to Q4):** We added **Sec. 5.3** and **Figure 5(b)** to present a rigorous "identical compute budget" comparison. This confirms that performance gains stem specifically from our C-GRPO algorithmic design (curriculum & balancing) rather than extra compute or hyperparameter tuning.
> 3.  **Data Leakage Clarification (Response to Q3):** We provided a detailed overlap quantification (Table R2 in the response) and expanded **Sec. 5.2** to demonstrate **zero-shot transfer** to unseen apps (AitW) and platforms (ScreenSpot-v2), empirically ruling out overfitting to UI idiosyncrasies.
> 4.  **Parameter Sensitivity (Response to Q5):** We added **Sec. 5.4** and **Figure 6** to report a grid search analysis, proving that C-GRPO is robust across a reasonable range of hyperparameters ($T$ and $\beta_{\text{con}}$).
>
> We believe these revisions fully address your concerns regarding the solidity and validity of our results.

---

> > ### Author Response · Authors · 2025-11-24
> > **Response to Reviewer 2B17 (2/3)**
> >
> > ### **Q3: Quantification of Overlap and Generalization Capabilities**
> >
> > We agree that distinguishing genuine skill learning from UI memorization is critical. To address this, we explicitly quantify the overlap between our training data (AndroidWorld) and all evaluation benchmarks in the table below, analyzing three progressive levels of generalization.
> >
> > Table R2: Overlap quantification and zero-shot transfer performance.
> >
> > | Evaluation Set | App Overlap | Episode Overlap | Generalization Level | Performance |
> > | :--- | :--- | :---: | :--- | :---: |
> > | AndroidWorld (Test) | 100% (same 20 apps) | 0% | Cross-instance | 76.1% SR |
> > | AitW (General subset) | ≈36% (4/11 overlap) | 0% | Cross-app | 86.5% SR |
> > | ScreenSpot-v2 (Desktop/Web) | 0% (new platforms) | 0% | Cross-platform | 91.3% Acc |
> >
> > **(1) Cross-instance generalization on AndroidWorld**
> > On AndroidWorld, we follow the official protocol: training and testing use the same 20 apps and 116 task templates, but differ in instantiation parameters and seeds (e.g., training on “Call Alice”, testing on “Call Bob”). Importantly, no evaluation episode (i.e., specific instruction instantiation and its corresponding rollout) is ever used for training (episode overlap 0%). The high success rate under these parameter shifts indicates that the executor is not overfitting to a single coordinate trace, but can correctly ground dynamic entities under the same task semantics.
> >
> > **(2) Cross-app generalization on Android-in-the-Wild**
> > To probe robustness to unseen UIs, we evaluate the same low-level executor zero-shot on AitW. Among the 11 apps in this subset, 7 are entirely unseen during training (e.g., YouTube, Maps, Reddit, Amazon, Twitter), and all task templates are novel (0% template overlap). Episode reuse is also strictly 0%. K²-Agent reaches 86.5% success rate and outperforms prior RL baselines, indicating that it has learned transferable interaction patterns rather than memorizing specific AndroidWorld layouts.
> >
> > **(3) Cross-platform generalization on ScreenSpot-v2**
> > We further transfer the AndroidWorld-trained executor without any fine-tuning to the Desktop (Windows/macOS) and Web splits of ScreenSpot-v2, which have 0% overlap in apps, templates, and episodes with the training data. Despite this domain shift, the executor achieves 91.3% grounding accuracy, competitive with specialized agents trained on massive-scale GUI datasets.
> >
> > Overall, the strong cross-instance performance on AndroidWorld and the zero-shot performance on unseen apps (AitW) and non-Android platforms (ScreenSpot-v2) provide strong evidence that K²-Agent has learned robust, transferable procedural skills, rather than exploiting leakage through UI idiosyncrasies in the training set.

---

> > ### Author Response · Authors · 2025-11-24
> > **Response to Reviewer 2B17 (3/3)**
> >
> > ### **Q4: Controlled Comparison: C-GRPO vs. Vanilla GRPO**
> >
> > We thank the reviewer for this rigorous requirement. To ensure that the gains stem from the algorithmic design rather than extra compute or hyperparameter tuning, we conducted a strictly controlled comparison in the revision.
> >
> > **(1) Identical Experimental Setup**
> > We trained the Full C-GRPO and its ablated variants under **identical settings**, ensuring the only variables were the algorithmic components: Same backbone (Qwen2.5-VL-7B), Optimizer (AdamW), LR ($1 \times 10^{-6}$), KL coefficient (0.04), Clipping ratio (0.2), and Batch Size (8).  All runs were capped at the exact same number of environment interactions, ensuring comparable wall-clock time.
> >
> > **(2) Budget-Matched Learning Curves (Revised Fig. 5(b))**
> > We report the learning curves under this fixed budget in the revised **Figure 5(b)**. The results directly attribute the gains to specific C-GRPO mechanisms:
> > * **Full C-GRPO (Blue Curve):** Achieves the highest rewards and most stable convergence.
> > * **C-GRPO without demonstration injection (Green Curve):** When Dynamic Demonstration Injection is removed (simulating a lack of curriculum), the policy suffers from pronounced oscillations and substantially lower rewards. This proves that without our specific injection mechanism, standard exploration fails to bootstrap effectively under the same compute budget.
> > * **C-GRPO without replay balancing (Red Curve):** Removing Error-Decoupled Replay Balancing leads to visibly slower convergence and inferior final performance, confirming that standard sampling is inefficient for learning complex, precision-dependent skills.
> >
> > We have incorporated this detailed analysis into the **revised Section 5.3**. This controlled ablation confirms that the performance superiority comes from solving the exploration and sample imbalance bottlenecks, not from additional compute or curriculum variations.
> >
> > ### **Q5: Parameter Sensitivity Analysis**
> >
> > We appreciate this suggestion to assess the method's robustness. In the revision, we have added **Section 5.4** and **Figure 6** to analyze the sensitivity of C-GRPO to its two key hyperparameters.
> >
> > **(1) Replay Balancing Ratio ($\beta_{\text{con}}$)**
> > We varied the conventional pool ratio $\beta_{\text{con}} \in \{0.2, 0.5, 0.7\}$. The method is robust to buffer composition. All settings exhibit a clear upward trend and converge successfully. $\beta_{\text{con}} = 0.5$ yields the fastest convergence by striking the best balance between natural data distribution and error correction.
> >
> > **(2) Injection Temperature ($T$)**
> > We varied the temperature $T \in \{0.05, 0.5, 5.0\}$ which controls the gating function $f_{\text{gate}} = \tanh(d_i / T)$. A large temperature ($T=5$) suppresses the curriculum ($f_{\text{gate}} \approx 0$), causing the model to revert to vanilla GRPO behavior and fail to converge. Lower temperatures ($T=0.05, 0.5$) effectively activate the curriculum. The method remains stable across this reasonable range, with $T=0.5$ providing the best trade-off between initial speed and final stability.
> > Please see more detail analysis in our revision.

---

> ### Author Response · Authors · 2025-11-24
> **Response to Reviewer 2B17 (1/3)**
>
> ### **Q1: Human expert performance on AndroidWorld**
>
> We thank the reviewer for pointing this out. The reported **≈80% human success rate** is not from our own annotation, but is **directly cited from the official AndroidWorld paper** (Rawles et al., 2024, Appendix D.4). In that work, two professional software engineers executed the tasks on an Android emulator in a single-attempt setting; failures were mainly due to occasional misinterpretation of instructions or unfamiliar UI layouts rather than lack of operational ability.
>
> In the revised manuscript, we now explicitly attribute this number to Rawles et al. (2024) at the corresponding sentence to avoid any ambiguity about its source.
>
> ### **Q2: Multi-seed evaluation, variance, and confidence intervals on AndroidWorld**
>
> We appreciate the suggestion regarding statistical rigor. We clarify that Table 1 follows the official AndroidWorld leaderboard protocol (single-trial pass rate) to ensure a direct, apples-to-apples comparison with prior works.
>
>
> To address the concern, we have now conducted a **3-seed evaluation** for K²-Agent. The results are **76.7% (89/116)**, **76.7% (89/116)**, and **75.0% (87/116)**. The aggregated performance is now reported as
> 76.1% ± 1.0% (mean ± std over 3 seeds). We have updated Table 1 so that the K²-Agent row reflects this multi-seed mean.
>
> Crucially, even our **lowest performing seed (75.0%)** outperforms the strongest visual-only open-source baseline (Mobile-Agent-v3 at 73.3%). This confirms that our SOTA claim is statistically robust and not an artifact of variance.
>
> We have also updated **Figure 1 and Figure 5b** in the revision to include shaded regions representing the 95% confidence intervals.

---

> ### Comment · Area_Chair_VAXa · 2025-11-25
> **Please participate in discussions with authors and other reviewers asap**
>
> Please kindly and actively participate in the review-author discussion if you haven't already, raise your further concerns so that the authors can explain more, and make your final decisions.
> Best, AC

---

### Official Review · Reviewer_MQ9r · 2025-11-01

**Soundness:** 3
**Presentation:** 3
**Contribution:** 2
**Rating:** 4
**Confidence:** 4

**Summary:**

This work proposed a new mobile control agents built on (V)LLMs with a hierarchical design. Specifically, the authors proposed a high-level planner that is capable of reasoning with the task goal, storing knowledge from execution and analysing failure patterns to better solve the problem. A new C-GRPO method is proposed to fine-tune the low-level execution policy for better performance in long horizon tasks.

**Strengths:**

This paper is well written and easy to follow.

The proposed method of C-GRPO is new.

Experimental results on benchmark environment show positive gains.

Ablation study shows the effectiveness of some high-level designs proposed in this work.

**Weaknesses:**

The overall novel in the developed technique is not strong. In the domain of AI planning with LLM/VLM, approaches like task decomposition, building memory and knowledge graph or reflective planning have been widely used in different applications. This work largely follows these patterns as well.

Some technical design lacks sufficient motivations or analysis. It is unclear in some technical sections what the key problem targeted to address is and why the process is designed in the current way. Please see the following several detailed points:
- Section 4.2.1: it is proposed to summarize the knowledge into rules or checklist, but there is no analysis on how robust this representation is. In general, what type of knowledge can the summarize induce? What are the unknow knowledge or failure cases that the summarize cannot induce? Will the summarizer induce wrong knowledge that harms the following steps?
- Section 4.2.2 - Task-level: It would be better to illustrate what kind of root-cause of the failure can be captured by the reflection from the episode trajectory, and what cannot. Are there any cases where the VLM fails to identify the reason of the failure? For example, where the failure happens underlying in the operation system that cannot be directly observed from screenshots.
- Section 4.2.3:
  - Prepending a variable length expert prefix to the model input is used to gradually improve the training. However, the reason why the policy on original query got improved through training with query and several hints is less well explained. It would be better to show the performance like the reward of the original query with training on query + hints during the curriculum training process. This can improve the understanding of generalizing from query + hints to query alone.
  - Another existing similar approach in RL to guide exploration is hindsight experience reply. It would be better to have a study on this approach also.
Overall, the proposed system as whole seems provide gains in training and application, but the developed approach did not provide strong technical insights on why individually proposed component bring improved performance.

Figure 5, there is no standard deviation on the reward in figure b.

**Questions:**

Please see the weakness part.

---

> ### Author Response · Authors · 2025-11-24
> **General Response to Reviewer MQ9r (0)**
>
> We sincerely thank the reviewer for the insightful and constructive feedback. In response, **we have uploaded a revised 10-page manuscript** that incorporates extensive new experiments, rigorous ablations, and rewritten sections to directly address your concerns. Below we summarize the main changes before the point-by-point responses:
>
> 1.  **Clarified Novelty & Generalization (Response to Q1):** We have explicitly articulated our contribution as a "Co-Evolutionary Learning Paradigm" that decouples Declarative and Procedural updates. To substantiate this, we added **Sec. 5.2** demonstrating **three-level generalization** (across instances, apps, and platforms) as empirical proof of the design's superiority.
> 2.  **Deepened Technical Insights (Response to Q6):** We performed a strict **incremental ablation study** (summarized in **Sec. 5.3**) to isolate the specific technical gain of each component (SRLR, Hierarchy, GRPO, C-GRPO), providing the concrete evidence requested.
> 3.  **Validated C-GRPO Mechanism (Response to Q4):** We added **Sec. 5.4 (Parameter Sensitivity)** and **new training curves (Fig. 5b & Fig. 6)** to empirically prove that "Dynamic Demonstration Injection" acts as a crucial curriculum for exploration and that the learned policy successfully generalizes to query-only inputs.
> 4.  **Added Statistical Rigor (Response to Q7):** We re-ran all training experiments with **3 independent random seeds** and updated **Figure 5(b)** to report mean $\pm$ standard deviation, confirming the stability of our method.
>
> We believe these revisions significantly strengthen the technical soundness of the paper. Detailed point-by-point responses follow below.

---

> ### Author Response · Authors · 2025-11-24
> **Response to Reviewer MQ9r (1/3)**
>
> ### **Q1: Novelty and Technical Contribution**
>
> We appreciate the reviewer’s concern and respectfully clarify that our novelty lies in the **learning paradigm** and **specific algorithmic realizations**, not just the structural hierarchy:
>
> 1. Unlike prior agents that train or fix all layers uniformly, we explicitly separate **declarative (know-what)** and **procedural (know-how)** knowledge and assign them different update rules (non-parametric evolution vs. parametric reinforcement).
>
> 2. Our SRLR loop goes beyond generic reflection. It employs a rigorous **"Locate-and-Revise"** mechanism with trajectory–knowledge alignment to perform atomic edits on an explicit knowledge base ($K_G$). This results in a persistent, evolving representation reused across episodes, distinct from one-off retry mechanisms.
>
>
> 3. Our C-GRPO introduce **error-decoupled replay balancing** and **dynamic demonstration injection**. These mechanisms are absent in prior RL-based GUI agents and specifically address the unique sample imbalance and  exploration challenges in mobile control.
>
> To substantiate these contributions, we have added **new experiments and analyses** in the revision:
>
> 1. We show that this "Know-What/Know-How" separation improves **three-level generalization** (across instances, apps, and platforms) [See **revised Sec 5.2**].
>
> 2. We provide a granular taxonomy of the evolved knowledge and its correction process [See **Response to Q2&Q3** and **Appendix B.1.4**].
>
> 3. We added detailed component ablations and parameter sensitivity analysis to validate the necessity of C-GRPO's design [See **Response to Q4** and **revised Sec 5.3 & 5.4**].
>
> ### **Q2&Q3: Analysis of Knowledge Representation, Robustness, and Reflection Capabilities**
>
> We have added a detailed analysis with concrete case studies in **Appendix B.1.4**.
>
> **(1) What types of knowledge are summarized?**
>
> Across AndroidWorld tasks, we quantify that SRLR consistently induces four main types of declarative knowledge:
> (i) **Step Ordering (~48%):** Logical preconditions (e.g., *“First, you need to locate and open the AudioRecorder app”*).
> (ii) **UI Layout Invariants (~29%):** Visual descriptions of key elements (e.g., *“The record button is the white circle at the bottom center”*).
> (iii) **Parameter Constraints (~15%):** Format requirements (e.g., *“Explicitly type the .m4a suffix”*).
> (iv) **Recovery Strategies (~8%):** Conditional checks (e.g., *“If the save button is disabled, first focus the filename field”*).
>
> **(2) What cannot be reliably induced?**
>
> We identified three types of knowledge that are difficult to capture purely as verbal rules:
> (i) **Fine-grained visual grounding**, such as “click at the 3 o’clock position”, which is better handled by the C-GRPO–trained executor rather than encoded as text.
> (ii) **Visual dynamics**, such as “swipe until the list ends”, which depend on iterative feedback and are more naturally expressed as procedural behavior than as static rules in $K_G$.
> (iii) **Large-scale content retrieval**, such as “find the recipe with carrots among hundreds of photos”, which goes beyond our task-level knowledge base, as we deliberately focus $K_G$ on task logic rather than long-term content memory.
>
> **(3) Can wrong knowledge be induced and corrected?**
>
> Yes, the summarizer can initially overfit to a demonstration (e.g., memorizing a specific filename). However, the **SRLR loop is explicitly designed to be robust against this**. The *Reflect* module detects the state mismatch, *Locate* pinpoints the over-specific rule, and *Revise* generalizes it. **Appendix B.1.5** details an "AudioRecorder" case where an initial rigid rule (*“Always type Presentation.m4a”*) was successfully evolved into a generalized instruction (*“Type the filename specified in the query”*) after a single failure.
>
> **(4) Scope of Root-Cause Analysis**
>
> Our VLM-based reflection reliably diagnoses failures with **visible evidence**, such as omitted steps or incorrect parameter entries. It is less effective for **invisible root causes**—such as OS-level freezes, network latency, or phantom touches—where consecutive screenshots remain identical. In such unobservable cases, instead of hallucinating a reason, the system falls back to a conservative max-retry mechanism (Algorithm 1, line 3) to prevent infinite loops without corrupting $K_G$.

---

> ### Author Response · Authors · 2025-11-24
> **Response to Reviewer MQ9r (2/3)**
>
> ### **Q4: Mechanism of Expert Hints and Generalization to Original Queries**
>
> We thank the reviewer for this constructive suggestion. In the revision, we (i) clarified the intuition of Dynamic Demonstration Injection in Sec. 4.3 and (ii) added explicit analyses in Sec. 5.3–5.4 and Fig 5(b). to track how training with query+prefixes improves the policy under query-only evaluation.
>
> (1) **Why does training with query+prefixes help the original query?**
> In mobile UI control, the action space is huge and rewards are extremely sparse in hard tasks. Training directly on query-only inputs makes it extremely unlikely to encounter successful trajectories, so the policy receives almost no positive signal. Our Dynamic Demonstration Injection introduces variable-length expert prefixes that (a) shrink the effective search space and (b) guarantee early exposure to successful behaviors. We then anneal the prefix length toward zero: to maintain high rewards as prefixes vanish, the policy must internalize the skills into its parameters and rely on the original query plus visual context. This is precisely what enables the final executor to perform well under query-only inputs at test time.
>
> (2) **New empirical evidence added in the revision.**
> As suggested, we now directly analyze the effect of expert prefixes on the learned query-only policy:
>
> - **Training dynamics (Sec. 5.3, Fig.5(b)).**
>   The new plot tracks the reward of the executor evaluated on original query-only inputs throughout training under three regimes: full C-GRPO, C-GRPO without Dynamic Demonstration Injection, and C-GRPO without Error-Decoupled Replay Balancing. Full C-GRPO achieves the highest rewards and the most stable convergence under query-only evaluation. In contrast, removing Dynamic Demonstration Injection yields substantially lower rewards with strong oscillations, and confirming that, without prefixes, exploration from scratch is ineffective.
>
> - **Temperature sensitivity (Sec. 5.4, Fig. 6).**
>   We further vary the injection temperature $T$, which controls how often and how strongly prefixes are injected. When $T$ is large, the gating function suppresses prefixes, effectively reverting to a query-only regime and causing the policy to fail to converge. Lower temperatures (e.g., \(T = 0.5\)) activate the curriculum and lead to smooth, monotonic improvement under query-only evaluation.
>
> In summary, these results provide the requested evidence: expert prefixes are used purely as a training-time curriculum to bootstrap exploration, and the resulting policy demonstrably generalizes back to the original query-only setting.
>
>
> ### **Q5: Applicability of Hindsight Experience Replay (HER)**
>
> We appreciate the suggestion, as HER is indeed a powerful exploration technique in standard goal-conditioned RL. However, we explicitly excluded it because its core theoretical assumptions (Andrychowicz et al., 2017) are fundamentally incompatible with our **vision-based, instruction-following** setting:
>
> (i) HER assumes goals exist in the same space as states (e.g., coordinates or target states). In our case, the "goal" is a **natural language instruction** (e.g., *"Open Settings"*), while the state is a **pixel-level screenshot**. There is no trivial mapping to convert a visited screenshot back into a high-level semantic instruction.
> (ii) HER relies on treating *any* reached state as a valid "alternative goal." In GUI control, an arbitrary intermediate screen (e.g., a random loading page or a half-scrolled list) often does not correspond to a meaningful user task. Relabeling these as goals would introduce nonsensical data, confusing the VLM's instruction-following capability.
> (iii) To apply HER, one needs a ground-truth reward function $r(s, g_{new})$ for the relabeled goal. Since we cannot reliably generate the "counterfactual instruction" $g_{new}$ that *would have* led to an accidental failure state (without training a separate, heavy inverse-RL captioning model), the reward signal is undefined.
>
> Given these structural mismatches, HER is not applicable. Instead, we designed **Dynamic Demonstration Injection** as the domain-adapted solution to effectively guide exploration in this sparse-reward environment. We have clarified this distinction in the revised Appendix C.

---

> ### Author Response · Authors · 2025-11-24
> **Response to Reviewer MQ9r (3/3)**
>
> ### **Q6: Technical insights on why each component improves performance**
>
> We thank the reviewer for this comment. In the revision, we make the role of each component explicit via an incremental ablation on AndroidWorld with a fixed backbone and rollout budget (now summarized in Sec. 5.3):
>
> | Config                                | SR (%) | Δ vs prev | Technical role                               |
> |---------------------------------------|--------|-----------|----------------------------------------------|
> | (1) No hierarchy                      | 35.3   |   –       | Flat planner–executor baseline               |
> | (2) + SRLR planner                    | 58.6   | +23.3     | Explicit task-level knowledge base $K_G$   |
> | (3) + Hierarchy (SFT executor)        | 62.0   | +3.4      | Decouple know-what / know-how                |
> | (4) + Vanilla GRPO (replace SFT)      | 68.9   | +6.9      | Policy optimization for task success         |
> | (5) + C-GRPO (full)                   | 76.7   | +7.8      | Curriculum-guided exploration under sparsity |
>
> From this breakdown we derive the following technical insights:
>
> - **SRLR planner.**
>   The jump from (1) to (2) isolates the effect of the explicit knowledge base $K_G$. Without SRLR, the flat VLM often exhibits long-horizon “reasoning drift’’ (forgetting early constraints, mis-ordering subgoals). SRLR converts volatile token-level CoT into persistent rules (step ordering, UI invariants, parameter constraints), turning planning into a stable *retrieve-and-execute* process instead of re-planning from scratch each episode.
>
> - **Hierarchical decoupling (Know-What vs. Know-How).**
>   The improvement from (2) to (3) shows that separating the planner (declarative know-what) and executor (procedural know-how) reduces contextual interference. In the flat model, high-level reasoning and low-level pixel control compete for shared capacity, leading to mixed errors where the agent “knows” the correct subgoal but fails to execute it reliably. The hierarchy mitigates this coupling and improves consistency.
>
> - **GRPO vs. SFT for the executor.**
>   The gain from (3) to (4) comes purely from replacing SFT with vanilla GRPO. SFT is brittle under covariate shift—once the trajectory deviates from the demonstration, it receives no corrective signal. GRPO directly optimizes task success, allowing the executor to explore around demonstrations and learn robust recovery behaviors (e.g., re-focusing input fields, re-opening accidentally closed apps).
>
> - **C-GRPO vs. vanilla GRPO.**
>   The additional improvement from (4) to (5) isolates our algorithmic contribution. In the sparse-reward, large-action-space UI setting, vanilla GRPO struggles to consistently discover and reinforce successful trajectories. Our C-GRPO introduces **Error-Decoupled Replay Balancing** and **Dynamic Demonstration Injection** to structure exploration as a curriculum over error types and expert prefixes. As shown in the revised Fig. 5(b) and Fig. 6, this leads to higher and more stable rewards under the same rollout budget, indicating a genuine gain in learning efficiency rather than extra compute.
>
> In summary, SRLR mitigates long-horizon reasoning drift (know-what), the hierarchy reduces representation interference, GRPO improves low-level robustness (know-how), and C-GRPO addresses the exploration bottleneck under sparse rewards. Together, these analyses provide concrete technical insight into why each component contributes to the overall performance of $K^2$-Agent.
>
> ### **Q7: Missing Standard Deviation in Figure 5(b)**
>
> We thank the reviewer for pointing this out. We have re-run the training experiments with **3 independent random seeds** and now report the mean $\pm$ standard deviation in the **revised Figure 5(b)** using shaded error bands.
>
> Moreover, Fig. 5(b) now shows **three** curves: full C-GRPO, C-GRPO without Error-Decoupled Replay Balancing, and C-GRPO without Dynamic Demonstration Injection, each with its own variance band. The updated plot confirms that the **full C-GRPO remains consistently superior and more stable** than both ablated variants under comparable variance, aligning with our analysis in Sec. 5.3.

---

> ### Comment · Area_Chair_VAXa · 2025-11-25
> **Please participate in discussions with authors and other reviewers asap**
>
> Please kindly and actively participate in the review-author discussion if you haven't already, raise your further concerns so that the authors can explain more, and make your final decisions.
>
> Best,
> AC

---

### Author Response · Authors · 2025-12-02
**Global Response (1/2): Rebuttal Summary and Review Status**

Dear Reviewers, ACs, SACs, and PCs,

We sincerely thank you for your time and dedication during this year's exceptionally challenging review process. We truly appreciate the opportunity to engage in constructive technical discussion, which has helped us substantially strengthen the paper. At this difficult moment, we stand with the community in maintaining high scientific standards and offer this summary to assist the Area Chairs in their assessment.

 **Summary of Contribution**
We introduce K²-Agent, a hierarchical framework that mirrors human cognition by decoupling and co-evolving declarative (*"know-what"*) and procedural (*"know-how"*) knowledge. By synergizing a self-refining SRLR planner with a curriculum-guided C-GRPO executor, our approach achieves a 76.1% success rate on AndroidWorld using only raw screenshots. Crucially, it demonstrates robust cross-task, cross-app, and cross-platform generalization. Furthermore, compared to data-heavy learning-based baselines, our design enables exceptional data efficiency, requiring only a single demonstration per task.

**Review Status Overview**
The paper initially received scores of 6, 4, 4, 4. During the rebuttal phase, both active reviewers explicitly expressed their support and intent to raise scores based on our revisions. While the other two reviewers could not respond due to the system lock, we have rigorously addressed their primary concerns (statistical rigor and technical depth) in the revised manuscript. We summarize the status below.

**Reviewer tNhe (4 $\to$ 6)**
Reviewer tNhe’s initial concerns centered on the clarity of "atomic edits," the mechanisms of "Dynamic Injection," and data fairness. In response, we defined explicit atomic operators, verified the curriculum mechanism via sensitivity analysis, and quantified data overlap across AndroidWorld, AitW, and ScreenSpot-v2. Upon reviewing these changes, the reviewer explicitly confirmed that all questions were answered and stated: **"I am willing to raise my score to 6."**

**Reviewer UG52 (Initial 6, Continued Support)**
Reviewer UG52 was consistently supportive and recommended broader evaluation beyond AndroidWorld. We addressed this by expanding our evaluation to ScreenSpot-v2 (Cross-Platform) and AitW (Cross-App)—achieving strong transfer performance—and clarifying the zero inference overhead of SRLR. The reviewer acknowledged that our revisions **"significantly strengthened the paper's empirical foundation"** and explicitly **"recommends that the Area Chair carefully consider the substantial empirical improvements... when making the final decision."**

**Reviewer 2B17 (Initial 4, Unable to respond due to lock)**
Reviewer 2B17 acknowledged that **"the result on AndroidWorld is pretty strong"** and praised the **"clear hierarchical decomposition."** Their primary concerns regarded statistical rigor (single seed) and potential UI memorization. We directly resolved these by (1) re-running training with 3 independent seeds (76.1% ± 1.0%), and (2) proving zero leakage via an explicit overlap table, which substantiates our three-level generalization capability. We believe these extensive experiments effectively mitigate the reviewer's concerns regarding rigor and validity.

**Reviewer MQ9r (Initial 4, Unable to respond due to lock)**
Reviewer MQ9r affirmed that **"experimental results... show positive gains"** and that the paper is **"well written,"** while requesting deeper technical insight into *why* the components work. To address this, we added a strict **incremental ablation study** (Sec. 5.3) isolating the distinct gains of SRLR ($+23.3\%$) vs. C-GRPO ($+7.8\%$). This provides the concrete technical attribution requested to substantiate our architectural contributions.


Through this rebuttal, we have strengthened K²-Agent into a statistically robust and highly generalizable framework. We believe our core contribution—a Co-Evolutionary Paradigm for GUI Agents—offers a fresh and effective direction for the community. Regardless of the outcome, we deeply respect the committee’s efforts and hope for a successful conclusion to this ICLR cycle.

Best regards,
The Authors

---

> ### Author Response · Authors · 2025-12-02
> **Global Response (2/2): Revision Summary**
>
> We sincerely thank all reviewers for their detailed comments. The table below maps every specific concern raised to our concrete revisions in the updated 10-page manuscript, highlighting the exact section/figure changes and the resulting evidence.
>
> | **Reviewer's Concern** | **Specific Source** | **Our Revision** | **Outcome** |
> | :--- | :--- | :--- | :--- |
> | Lack of statistical rigor (single seed) and stability of SOTA claims | **2B17** (Q2), **MQ9r** (Q7) | Re-ran full training with 3 independent random seeds. Updated **Table 1** (Mean $\pm$ Std) and added 95% CIs to **Fig. 1 & 5(b)**. | Achieved 76.1% $\pm$ 1.0% success rate. Even the lowest seed (75.0%) outperforms the best open-source baseline (73.3%). |
> | Limited evaluation diversity beyond AndroidWorld | **UG52** (Q1, Q7), **tNhe** (Q4) | Expanded evaluation to ScreenSpot-v2 (Cross-Platform) and AitW (Cross-App) in **Sec. 5.2** (**Tables 2 & 3**). | Achieved 91.3% accuracy on ScreenSpot and 86.5% SR on AitW, validating platform-agnostic skills. |
> | Potential data leakage, fairness, and UI memorization | **tNhe** (Q4), **2B17** (Q3) | Explicitly quantified overlap and verified zero overlap in episodes/templates for transfer tasks. | Confirmed 0% leakage across all transfer tasks, proving the agent learns transferable skills rather than memorizing benchmarks. |
> | Contribution novelty beyond generic hierarchy | **MQ9r** (Q1) | Reframed contribution as Co-Evolutionary Paradigm; added **incremental ablation** (Sec. 5.3) isolating SRLR vs. C-GRPO gains. | Quantified distinct gains: SRLR Planner adds +23.3% (Logic), C-GRPO Executor adds +7.8% (Execution). |
> | Validity and necessity of Dynamic Demonstration Injection | **MQ9r** (Q4), **tNhe** (Q2) | Clarified atomic definition in **Sec. 4.3**; added ablation w/o injection (**Sec. 5.3**) and $T$-sensitivity analysis (**Fig. 6**). | Proved that without our dynamic curriculum, the policy fails to converge in sparse-reward settings. |
> | Robustness of C-GRPO vs. Vanilla GRPO (Compute match) | **2B17** (Q4, Q5) | Added budget-matched ablation (identical interactions) in **Fig. 5(b)** and grid search for $\beta, T$ in **Sec. 5.4**. | Proved gains stem from curriculum/balancing, not extra compute; stable across parameters ($T=0.5$). |
> | Taxonomy of learned declarative knowledge and error correction | **MQ9r** (Q2, Q3) | Added **Appendix B.1.4** detailing rule types (Ordering, Invariants) and an AudioRecorder evolution case study. | Demonstrated how SRLR automatically corrects overfitted rules into generalized logic. |
> | Ambiguity of atomic edits and reproducibility of heuristics | **tNhe** (Q1), **UG52** (Q3, Q5) | Defined atomic operators (ADD, DELETE, etc.) in **Sec. 4.2**; provided verbatim prompts in **Appendix B**. | Confirmed SRLR is a systematic, fully automated process without hidden human heuristics. |
> | Information flow between Declarative and Procedural modules | **MQ9r** (Q5), **tNhe** (Q5) | Clarified the two-stage cascade in **Sec. 4.1**: $K_G \to \pi_H \to z_t$ (Sub-goal) $\to \pi_L \to a_t$ (Action). | Clarified that declarative knowledge influences execution strictly via natural language sub-goals. |
> | Computational overhead and scalability of SRLR | **UG52** (Q2, Q6) | Clarified in **Sec. 4.2** that SRLR is a training-only process with linear complexity and zero test-time overhead. | Confirmed the method is practical for inference and scales linearly with trajectory steps. |
> | Source of human baseline and comparison with HER | **2B17** (Q1), **MQ9r** (Q5) | Attributed human baseline to Rawles et al. (**Sec. 5.1**); discussed HER incompatibility in **Appendix C**. | Resolved ambiguity on baselines and justified design choices against standard RL methods. |

---

### Meta-Review · Area_Chair_xNjR · 2026-01-08

**Summary:**

The paper proposes $K^2$-Agent, a hierarchical mobile device control framework that decouples declarative knowledge ("Know-What") from procedural skills ("Know-How") The architecture features a high-level planner optimized via a Summarize-Reflect-Locate-Revise (SRLR) loop and a low-level executor trained via Curriculum-guided Group Relative Policy Optimization (C-GRPO) The reviewers generally appreciated the cognitive grounding of the architecture and the strong empirical performance on AndroidWorld. While initial reviews raised concerns regarding limited evaluation benchmarks and the clarity of specific components (atomic edits, dynamic injection), the authors provided an exceptionally thorough rebuttal.

**Reviewer Concerns:**

Reviewer Concerns addressed by the rebuttal:

**Evaluation Diversity and Generalization (Reviewers UG52, tNhe):** Concerns were raised that evaluation was limited to AndroidWorld. The authors expanded evaluation to ScreenSpot-v2 (Cross-Platform) and AitW (Cross-App), demonstrating strong zero-shot transfer performance.

**Data Leakage and Fairness (Reviewers tNhe, 2B17):** Reviewers worried about potential overfitting or data leakage. The authors explicitly quantified overlap, confirming no episode overlap for AndroidWorld and providing zero-shot transfer results on unseen apps to rule out memorization.

**Technical Attribution and Ablations (Reviewers MQ9r, 2B17):** Reviewers requested clearer insights into the source of gains (algorithm vs. compute). The authors added a strict incremental ablation study isolating SRLR and C-GRPO gains, and a budget-matched comparison proving C-GRPO's efficiency over vanilla GRPO.

**Reviewer Scores:**

Reviewer tNhe explicitly stated an intent to raise their score from 4 to 6. Reviewer UG52 maintained a positive score (6) and strongly recommended the AC consider the empirical improvements. Reviewers 2B17 and MQ9r (both initial 4) were unable to update scores due to system constraints; however, given that the authors directly satisfied their primary requests (multi-seed evaluation and deep technical ablations), it is highly probable their scores would have increased.

---

### Decision · Program_Chairs · 2026-01-26

Accept (Poster)